# Recombinant African Swine Fever Virus Arm/07/CBM/c2 Lacking CD2v and A238L Is Attenuated and Protects Pigs against Virulent Korean Paju Strain

**DOI:** 10.3390/vaccines10121992

**Published:** 2022-11-23

**Authors:** Daniel Pérez-Núñez, Sun-Young Sunwoo, Raquel García-Belmonte, Chansong Kim, Gonzalo Vigara-Astillero, Elena Riera, Dae-min Kim, Jiyun Jeong, Dongseob Tark, Young-Seung Ko, Young-Kook You, Yolanda Revilla

**Affiliations:** 1Microbes in Health and Welfare Department, Centro de Biología Molecular Severo Ochoa, CSIC-UAM, c/Nicolás Cabrera 1, 28049 Madrid, Spain; daniel_perez@cbm.csic.es (D.P.-N.); raquel.g.b@cbm.csic.es (R.G.-B.); gvigara@cbm.csic.es (G.V.-A.); elena.riera@cbm.csic.es (E.R.); 2Careside Co., Ltd., Sagimakgol-ro 45 Beongil 14, Seongnam-si 13209, Gyeonggi-do, Republic of Korea; sunwoosy@gmail.com (S.-Y.S.); messiah383@naver.com (C.K.); wldbs3290@konkuk.ac.kr (J.J.); careside@careside.co.kr (Y.-K.Y.); 3Laboratory for infectious Disease Prevention, Korea Zoonosis Research Institute, Jeonbuk National University, 79 Gobong-ro, Ma-dong, Iksan 54531, Jeollabuk-do, Republic of Korea; daeminkk@gmail.com (D.-m.K.); tarkds@jbnu.ac.kr (D.T.); dudtmd3315@naver.com (Y.-S.K.)

**Keywords:** ASFV, vaccine, CD2v, A238L, genotype II, safe, protection

## Abstract

African swine fever (ASF) is an obligated declaration swine disease, provoking farm isolation measures and the closing of affected country boarders. ASF virus (ASFV) is currently the cause of a pandemic across China and Eurasia. By the end of 2019, ASF was detected in nine EU Member States: Bulgaria, Romania, Slovakia, Estonia, Hungary, Latvia, Lithuania, Poland and Belgium. The affected area of the EU extended progressively, moving mostly in a southwestern direction (EFSA). Inactivated and/or subunit vaccines have proven to fail since certain virus replication is needed for protection. LAVs are thus the most realistic option, which must be safe, effective and industrially scalable. We here generated a vaccine prototype from the Arm/07/CBM/c2 genotype II strain, in which we have deleted the EP402R (CD2v) and A238L genes by CRISPR/Cas9 in COS-1 cells, without detectable further genetic changes. The successful immunization of pigs has proven this vaccine to be safe and fully protective against the circulating Korean Paju genotype II strain, opening the possibility of a new vaccine on the market in the near future.

## 1. Introduction

African swine fever virus (ASFV), the sole member of the Asfarviridae family [1,2,3], is the etiological agent of African swine fever (ASF), a serious disease affecting both wild boar and domestic pigs. An outbreak in the Caucasus in 2007 started its spread across Russia and Eastern Europe, currently affecting Poland, Ukraine, Latvia, Bulgaria, Hungary, Moldova, Romania, Serbia and Slovakia; an outbreak detected in Belgium in 2018 was threatening the French border and produced the first case detected in wild boar in Germany during 2020 [4]. Importantly, in August 2018 the first ASFV outbreak was reported in China [5], one of the most important pork producer countries in the world, and since then, the virus has spread all over China and is currently affecting to neighboring countries as Vietnam, Laos, Myanmar, Korea and Philippines. Very recently, an outbreak was also declared in the Dominican Republic and Haiti during 2022 [6], being the first report of ASFV outbreak in the Americas in the last 30 years. The situation is economically dramatic, and a serious unbalance of the food chain could soon happen in the world; consequently, ASFV now represents one of the most important social, environmental, swine exploitations and food industrial concerns worldwide.

Despite efforts to develop an effective vaccine against ASFV [2,7,8], only control and eradication measures have been implemented so far, based on the early diagnosis, regionalization and eradication. However, the rapid spread of the disease shows these measures as insufficient to control the current pandemic situation, and the development of a vaccine is urgently required. Currently, only one vaccine prototype, based on a live attenuated vaccine (LAV), is being implemented in Vietnam [9], the effects and effectiveness of which are being carefully analyzed.

The most realistic approach for ASF vaccine development is the generation of LAVs, which consist of either naturally attenuated isolates or attenuation of virulent strains via cell passage or genetic modification. Currently, several LAV candidates have been developed showing degrees of protection against heterologous and/or homologous challenges, from 60 to 100%, having different side effect profiles in vaccinated animals [10,11,12,13,14,15,16,17,18,19,20,21], reviewed in [22].

Vaccine development against ASFV based on LAV prototypes is already realistic, although there are certain aspects to be taken into account for industrial development. On the one hand, the use of simply naturally attenuated strains is not admissible, both because of their safety problems and the lack of differentiating infected from vaccinated animals (DIVA). On the other hand, the vast majority of prototypes have been made in porcine primary cells, such as macrophages or PBMCs, which is not industrially applicable due to cost, lack of reproducibility and ethical issues. Therefore, a suitable cell line to generate and grow the cell prototype is essential for the development of LAV against ASFV. Finally, not all prototypes have been properly genetically analyzed so far, rather focusing on analyzing the lack of the deleted gene(s) and not on the full genome sequence [12,19,20]. Sequencing based on next-generation sequencing (NGS) studies of sufficient quality (high mean coverage) are necessary for the correct characterization of the prototypes, as well as for the study of their genetic stability for industrial development.

Here we present a new candidate LAV against ASFV based on the virulent genotype II Arm/07/CBM/c2 strain [23], to which we have deleted two genes, EP402R (encoding for CD2v) and A238L, using CRISPR/Cas9 technology in COS-1 cells. A238L gene is a homolog of the NFkB inhibitor IkB [24,25], which is involved in the regulation of the immune response, mainly through the control of NFkB [26] and NFAT by inhibiting calcineurin phosphatase activity [27]. Our group previously described that A238L is able to inhibit inflammatory regulators such as TNF-α [28], cyclooxygenase-2 (COX2) [29] and nitric oxide synthase (NOS) [30] through inhibition of the transcriptional co-activator p300 [31,32]. On the other hand, EP402R encodes for the CD2v protein, which is involved in hemadsorption [33,34], binding of different host proteins [35,36], and has been shown to induce IFN-I secretion via NFkB and apoptosis in vitro [37]. In certain cases, CD2v has been linked to virulence in vivo, although this issue is controversial. In some prototypes, deletion of CD2v from an ASFV virulent strain has had a direct or partial effect on virulence [16,38,39], while in other cases its deletion had no in vivo effect compared to the parental virulent strain [40,41]. Thus, it appears that its role in virulence seems to depend at least in part on the context of the viral genotype. In this regard, very recently, as this manuscript was being written, a LAV prototype based on genotype IX in which EP402R and A238L were deleted proved to be attenuated, but failed to confer adequate protection [10].

In this study, we have succeeded in deleting these two genes generating ArmΔCD2v-ΔA238L LAV prototype. Genetic analysis by high-mean-coverage NGS revealed no extra deletions and very few unexpected point mutations that mainly dropped in intergenic areas. In vitro analyses showed that the mutant was able to grow similarly to the wild-type in porcine macrophages, and that it shows an attenuated phenotype in terms of IFN production and the activation of the cGAS-STING pathway, which led us to consider it as a potential in vivo vaccine prototype. Indeed, in vivo studies demonstrated that the prototype is fully attenuated, without any apparent side effect, and is able to protect 100% against the virulent Korean Paju strain. Consistent with the protection, a high production of antibodies is observed early after vaccination and from the beginning of the challenge, which was maintained throughout the experiment. Finally, we found no or very low values of viremia in blood, oral and fecal fluids before and after the challenge, and this happened in very low amounts and only in some tissues analyzed.

All these results, added to the fact that the generation and growth of the vaccine prototype has been carried out in a cell line, which is optimal for an industrial scale-up, make this LAV a potential and promising vaccine against ASFV and could soon become an important tool in the fight against this disease worldwide.

## 2. Materials and Methods

### 2.1. Design and Generation of CRISPR-Cas9 Vectors for the Deletion of EP402R and A238L Genes from the Virulent African Swine Fever Virus (ASFV) Strain Arm/07/CBM/c2

Two types of vectors were used: (i) one derived from pSpCas9(BB)-2A-Puro (PX459), in which nuclear localization sequence (NLS) has been deleted (pSpCas9(BB)ΔNLS-2A-Puro); and (ii) a pcDNA3.1-derived vector containing the flanking sequences of the target genes (EP402R or A238L), surrounding the marker gene EGFP derived from the pEGFP-E3 vector, or, alternatively, the mCherry gene derived from pmCherry-N1.

For the pSpCas9(BB)ΔNLS-2A-Puro vectors, specific gRNAs were cloned to interrupt either the EP402R gene, coding for the CD2v protein, or the A238L gene. We designed the gRNA based on the sequence of the ASFV strain Arm/07/CBM/c2 (LR812933.1) using Protospacer. The designed gRNA sequences were as follow: 5′-TCTTCATTAGATTCAGGTGG-3′ and 5′-GCTAGCTACATGTGGAAAAGC-3′ for EP402R gene, and 5′-GCATATCCACATGAATACCGAGG-3′ and 5′-GAATTTTTTAAAACAGATCCGTGG-3′ for A238L gene, which were cloned into the pSpCas9(BB)ΔNLS-2A-Puro vector following the procedure described in [42], generating the following vectors: pSpCas9(BB)ΔNLS-2A-Puro_EP402R-gRNA-0 and pSpCas9(BB)ΔNLS-2A-Puro_EP402R-gRNA-8 and pSpCas9(BB)ΔNLS-2A-Puro_A238L-gRNA-1 and pSpCas9(BB)ΔNLS-2A-Puro_A238L-gRNA-2.

For the generation of pcDNA-derived vectors, we used a pcDNA3.1 vector as a backbone in which we first cloned the EP402R gene and its flanking regions (500 bp upstream and 500 bp downstream) (73,362–75,716), generating the pFL-EP402R vector. For that, we designed specific probes to clone the EP402R and flanking regions by In-Fusion technology (Clontech), which were as follow: 5′-CCAGATATACGCGTTGTTTGAAAAAAAAAATAGATGATTATAGTATATTAATAATTGG-3′ and 5′-TTTCCGCCTCAGAAGTTTTGGGAACTGTGGGCCTC-3′; and 5′-AACGCGTATATCTGGCCCG and CTTCTGAGGCGGAAAGAACCA-3′. PCR were performed using Phusion High-Fidelity PCR Master Mix with HF Buffer (ThermoScientific, Waltham, MA, USA) and purified viral DNA from ASFV Arm/07/CBM/c2 strain [23]. Once the pFL-EP402R vector was obtained, it was used as a backbone for the generation of the next vector, pFL-ΔEP402R-GFP. For that, we designed specific probes for the insertion of the EGFP gene into the pFL-EP402R vector, eliminating the EP402R gene, by In-Fusion technology. The probes were as follow: 5′-TATGTACTATATATTAATTATTTAACCTTTCAAGC-3′ and 5′-TTTATGAACATATGTTTTATAATATAGTATCAAAAAC-3′; 5′-ACATATGTTCATAAAGACATTGATTATTGACTAGTTATTAATAG-3′ and 5′-AATATATAGTACATACCATAGAGCCCACCG-3′. As the next step, we substituted the CMV promoter by an ASFV promoter (p72 promoter, described in [43]). For that, we designed the specific probes: 5′-TGGAGTTCCGTATTTAATAAAAACAATAAATTATTTTTATAACATTATATAGGTCGCCACC-3′ and 5′-GGTGGCGACCTATATAATGTTATAAAAATAATTTATTGTTTTTATTAAATACGGAACTCCA-3′ to amplify the p72 promoter; and 5′-GGTCGCCACCATGGTGAG-3′ and 5′-CGGAACTCCATATATGGGCTATG-3′ to linearize the pFL-ΔEP402R-GFP vector by eliminating the CMV promoter. This product was cloned into the pFL-ΔEP402R-GFP by In-Fusion technology, generating the final vector, pFL-ΔEP402R-p72GFP.

Regarding the A238L gene, we used a pcDNA3.1 vector as a backbone in which we clone the A238L gene and its flanking regions (500 bp upstream and 500 bp downstream) (50,730–52,410), generating the pFL-A238L vector with the same technology. For that we used the following probes: 5′-CTTCTGAGGCGGAAAGAACCA-3′ and 5′-AACGCGTATATCTGGCCCG-3′; 5′-CCAGATATACGCGTTGAAAACCTGCTGTTCTGATAAGAACA-3′ and 5′-TTTCCGCCTCAGAAGAGGATTAGATGCGACGCC-3′. Once the pFL-A238L vector was obtained, it was used as a scaffold for the generation of the next vector, pFL-ΔA238L-p72-mCherry. For that, we used the following probes: 5′-TTTGAATGTCGCCCCTACTCC-3′ and 5′-AAAAGCTTAACAAGTATGGAAAGTAATCTCTCA-3′, 5′-ACTTGTTAAGCTTTTGACATTGATTATTGACTAGTTATTAATAGTAATCAATTACGG-3′ and 5′-GGGGCGACATTCAAATAAGATACATTGATGAGTTTGGACAAACC-3′, substituting the A238L gene by mCherry gene under p72 promoter, from a pFLΔEP402R-p72-Cherry previously generated in our laboratory.

### 2.2. Generation of Recombinant Virus Arm-ΔCD2v-GFP-ΔA238L-Cherry by CRISPR-Cas9 Technology

The recombinant virus was generated in COS-1 cells, from the American type culture collection (ATCC), grown in Dulbecco-modified Eagle medium (DMEM) supplemented with 2 mM l-glutamine, 100 U/mL gentamicin, nonessential amino acids and 5% fetal bovine serum. Cells were grown at 37 °C in a 7% CO_2_ atmosphere saturated with water vapor. We first generated the Arm-ΔCD2v-GFP virus, and after purification and further genetic characterization, we used the Arm-ΔCD2v-GFP virus as a backbone for the generation of the Arm-ΔCD2v-GFP-ΔA238L-Cherry (Arm-ΔCD2v-ΔA238L).

COS-1 cells (seeded a six-well plate at 90% confluence) were co-transfected with specific pSpCas9(BB)ΔNLS-2A-Puro gRNA (2 µg) together with the respective pcDNA-derived vector (2 µg) with FuGene HD (Promega, Madison, WI, USA). Additionally, 24 h post transfection, puromycin (Sigma, Saint Louis, MO, USA) was added to the media of transfected cells (1 µg/mL). After 48 h, transfected cells were infected at two different MOI (1 and 0.1) with Arm/07/CBM/c2 ASFV strain to generate Arm-ΔCD2v-GFP, or with Arm-ΔCD2v-GFP to generate Arm-ΔCD2v-ΔA238L. At five days post infection (dpi) cell and medium was collected and conserved at −80 °C.

### 2.3. Isolation of Recombinant Viruses from Wild-Type Viruses by Plaque Isolation

The collected recombinant virus was used to infect COS-1 cells. After 1 h 30 min of viral adsorption, the inoculum was removed and DMEM 2% fetal bovine serum with carboxymethylcellulose (CMC) was added. After 4–7 dpi viral plaques appeared and were identified by fluorescent microscopy. Recombinant plaque was collected by sterile tips in 40 µL of DMEM and conserved at −80 °C. After three freeze/thaw cycles, the extracted virus was used to infect new COS-1 cells by using the same procedure. This plaque isolation method was repeated at least five times in order to isolate the recombinant apart from the parental virus.

During the isolation procedure, wild-type or parental contaminant detection was checked by PCR. For that, 10 µL of the isolated plaque is digested with proteinase K (Sigma, Saint Louis, MO, USA) in 1.5 mM MgCl_2_, 50 mM KCl, 0.45% Tween20, 0.45% NP40 and 10 mM TrisHCl pH 8.3 buffer, incubated 30 min at 45 °C and then 15 min at 95 °C to inactivate the proteinase K. The digested isolated plaque was used as a DNA template for PCR to detect the presence of the recombinant and parental virus. The oligos used are listed in Table 1.

When no parental virus was detected by PCR, the recombinant virus was grown by infecting six to eight P100 plates of COS-1 cells. After 3 dpi, the total virus was collected and subjected to freeze–thaw cycles. After centrifugation at 3000 rpm 5 min at RT, supernatant was collected and centrifuged at 7000 rpm o/n at 4 °C. The pellet was resuspended in fresh DMEM medium and a 10 µL sample was collected in order to check for parental virus contamination by PCR, as explained above. Further dilution limit techniques may be required to fully purify the recombinant virus without parental contamination. When no contamination was detected, the virus was grown for DNA extraction and next generation sequencing (NGS) analysis.

### 2.4. Viral DNA Extraction for NGS Analysis

Viral DNA extraction of the recombinant virus for NGS analysis was performed as described previously [23]. Briefly, recombinant virus was grown in six to eight P100 pre-confluent COS-1 cells. After 3–4 dpi, supernatant containing extracellular virions was collected and centrifuged at 7000 rpm o/n at 4 °C. The pellet was resuspended in cold and filtrated 10 mM Tris-HCl pH 8.8. and treated with 0.25U/µL DNAse I (Sigma, Saint Louis, MO, USA), 0.25 U/µL Nuclease S7 (Sigma, Saint Louis, MO, USA) and 20 µg/mL RNAse A (Promega) in 800 mM Tris-HCl pH 7.5, 200 mM NaCl, 20 mM CaCl_2_ and 120 mM MgCl_2_ during 2 h at 37 °C, and further incubation with 12 mM EDTA (Sigma, Saint Louis, MO, USA) and 2 mM EGTA (Sigma, Saint Louis, MO, USA) 10 min at 75 °C. After that, the solution was treated with 200 µg/mL proteinase K (Sigma, Saint Louis, MO, USA) in 0.5% SDS for 1 h at 45 °C. Next, viral DNA was precipitated incubating 1:1 volume of the sample with phenol:chloroform:isoamilic acid at 25:24:1. After centrifugation at 10,000 rpm for 3 min at RT, watery fraction was transferred and further incubated with: 0.1 volumes of 3M acetic acid pH 5.2; 1 µL LPA (Sigma, Saint Louis, MO, USA) and 2 volumes of cold 100% ethanol for 1h at −80 °C. Then, the sample was centrifuged at 13,000 rpm for 30 min at 4 °C and the supernatant was discarded. The pellet was washed ones with cold 70% ethanol and dry on air. Finally, the pellet was resuspended in 10 mM Tris pH 8.8.

### 2.5. Illumina Sequencing and Data Analysis

Viral DNA was submitted to MicrobesNG (Birmingham, UK). Illumina libraries were prepared with NEBNext ultra DNA library prep kit (New England Biolabs, Ipswich, MA, USA). The DNA sample was fragmented in a Covaris instrument and sequenced on an Illumina MiSeq device as paired-end (2 × 250 bp) reads. Illumina reads from each sequenced sample were trimmed using Trimmomatic v0.36 [44] and quality-filtered (QF) with PrinSeq v1.2 [45]. Only paired QF reads were considered for further analysis. These paired QF reads as referred as Illumina QF reads in the text. Then, resulting paired reads were aligned against the Arm/07/CBM/c2 reference sequence (accession number LR812933.1), by using Bowtie 2 v2.3.4.1 [46] with default parameters.

Alignment files were then used for the variant calling process with GATK (v4.1.2). Numbers of single nucleotide polymorphisms (SNPs) and indels were determined and characterized by their location in coding and non-coding regions. Genetic variants were annotated using SnpEff [47] software (v5.0c). Shortly, a library was generated according to the SnpEff manual using the .gbk file from Arm/07/CBM/c2 reference sequences obtained from Genbank. VCF files generated by GATK were annotated using both libraries and the resulting annotated variants were analyzed using Libreoffice Calc. To confirm the mutations identified within the coding regions (SNPs and InDels), we amplified by PCR these regions by Phusion High-Fidelity PCR Master Mix (ThermoFisher, Waltham, MA, USA) and were then subject to Sanger sequencing (Macrogen, Seoul, Republic of Korea). The following primers were used: 5′-CCCTTAATTTGTTTTAACAAATATTATAACATCTAAG-3′ and 5′-GAATGTGTAAA AATAAAGTGTTTAGTGACC-3′ for ASFV G ACD 00190; 5′-GATTT AAACCCGGCTGAGATAGCC-3′ and 5′-GGACACAACATGAAG GTTCTAGG-3′ for MGF110-14L; and 5′-GTAAGCCGA AGGAAGACGAGTC-3′ and 5′-CATAGTACATCCGCTCAATATATTAAATGAAGG-3′ for D205R.

A coverage plot representing Arm-ΔCD2v-ΔA238L Illumina reads mapped against Arm/07/CBM/c2 reference genome was generated. Shortly after, the read coverage values of the Arm-ΔCD2v-ΔA238L were calculated using BEDtools software (v.2.27.1) and plotted using Rstudio software (v.2022.02.3+492).

The de novo assembly of Illumina reads was generated with Unicycler [48], which is an assembly pipeline that works as a SPAdes optimizer to assemble Illumina-only read sets, was used. It tries assemblies at a wide range of k-mer sizes, evaluating each graph and choosing the one that best minimizes both contig and dead ends. With the purpose of improving the process Scaffold_builder [49] software was used to order those contigs generated by draft sequencing along a reference sequence where gaps are filled with N’s and small overlaps are aligned with Needleman–Wunsch algorithm. Due to the presence of gaps in the generated assembly, Gapfiller [50], a tool that calculates and closes the size of the gaps by using the distance information derived from the paired read data, was used. Finally, to extend the extremes ends of the viral chromosome assembly, a consensus sequence based on the reads mapped against the reference genome was generated. For that, an in-house script that combines bcftools and seqtk to perform variant calling and quality filter (Q < 20), was created. We finally obtained a single contig of 189,266 bp.

### 2.6. Viral Growth Kinetics

Porcine alveolar macrophages (PAMs) were seeded in M12 plates (1.5 × 10^6^ cells/well) and mock or infected with Arm07/CBM/c2 (WT) or Arm-ΔCD2v-ΔA238L (0.5 PFU/cell) in DMEM–10% porcine serum. At 0, 24, 48 or 72 h post-infection (hpi) the cells were collected and subjected to three freeze–thaw cycles. Total virus was titrated in COS-1 cells by using a TCID-50 assay. Briefly, COS-1 cells were seeded in p96 plates (7000 cells/well) and infected with several dilutions of each sample. After 72 hpi cells were fixed (PFA 4%), and in the case of Arm/07/CBM/c2, permeabilized with 0.2% Triton X-100, and a TCID50 titration assay was performed using green (GFP) and red (mCherry) fluorescence for Arm-ΔCD2v-ΔA238L or using viral p32 staining using anti-p32 primary antibody (S-5C1) (1/500) and anti-mouse Alexa Fluor 488 secondary antibody (A-21206) from Invitrogen for Arm/07/CBM/c2. Biological duplicates were used.

### 2.7. RT-qPCR Assay

PAM were seeded in p60 plates (6·× 10^6^ cells/plate) and mock or infected with Arm07/CBM/c2 (WT) or Arm-ΔCD2v-ΔA238L (3 PFU/cell) in DMEM–10% porcine serum. At 4, 8 or 16 hpi, the total RNA was harvested from cells using a RNeasy kit (Qiagen, Hilden, Germany). cDNA was synthesized using a NZY first-strand cDNA synthesis kit (NZYTech, Lisbon, Portugal). qPCR was performed using an CFX384 touch real-time PCR detection system (Bio-Rad, Hercules, CA, USA) with SYBR green master mix (NZYTech). Gene expression levels were normalized to the housekeeping gene (18S rRNA), and these values were relative to the mock values. The primers used were 5′-GGCCCGAGGTTATCTAGAGTC-3′ and 5′-TCAAAACCAACCCGGTCA-3′ for porcine 18S rRNA detection, 5′-GTGGAACTTGATGGGCAGAT-3′ and 5′-TTCCTCCTCCATGATTTCCTC-3′ for porcine IFN-β detection, 5′-ACTTCCTAAGCCTTACAGTCGT-3′ and 5′-AGTGGTTGTGTTGAGGGACG-3′ for viral EP402R detection and 5′-GCAGATCCGACTCAAAAAGACT-3′ and 5′-ACTCCATATTTCCTGTAAAGACTGC-3′ for viral A238L detection. Biological and experimental triplicates were used.

### 2.8. Determination of cGAS-STING Pathway Activation by Western Blot

PAMs were seeded in M6 plates (3× 10^6^ cells/well) and mock or infected with Arm07/CBM/c2 (WT), Arm-ΔCD2v-ΔA238L, NH/P68, Ug-AT or E70 at the indicated MOI in DMEM–10% porcine serum. At the indicated times, the cells were collected, washed with PBS and lysed with a radioimmunoprecipitation assay (RIPA, Tris-HCl 50 mM, NaCl 150 mM, Triton 1%, Deoxycholate 0.5%, SDS 0.1%, produced in-house) buffer supplemented with protease and phosphatase inhibitors (Roche, Basel, Switzerland) for 30 min at 4 °C. Then, lysates were sonicated and centrifuged at 13,000 rpm for 10 min at 4 °C. The supernatants were collected and equal amounts of protein were used. Samples were resolved by sodium dodecyl sulfate polyacrylamide gel electrophoresis (SDS-PAGE, BioRad, Hercules, CA, USA) and transferred to Immobilon-P membranes (Millipore, Burlington, MA, USA). The membranes were incubated with the following specific primary antibodies: anti-pSTING (Ser366, catalog No. 85735), anti-pTBK1 (Ser172, catalog No. 5483) and anti-pIRF3 (Ser396, catalog No. 4947) were purchased from Cell Signaling, Danvers, MA, USA; anti-actin antibody (Sc-47778) from Santa Cruz Biotechnology, Dallas, TX, USA (1/5000); and anti-p32 antibody (S1D8) for viral protein p32 (1/5000) and anti-p72 antibody for viral protein p72 (1/1000) were produced in-house; diluted in Tris-buffered saline (TBS) supplemented with 1% milk. Membranes were washed three times with TBS and exposed 1 h to specific peroxidase-conjugated secondary antibodies: anti-rabbit and anti-mouse immunoglobulin G coupled to peroxidase (1/5000 and 1/2000, respectively) from Amersham Biosciences and anti-mouse-IgGκ secondary antibody (1/1000) from Santa Cruz Biotechnology. Chemiluminescence detection was performed using ECL Prime (Amersham Biosciences, Amersham, UK). Densitometry was performed with ImageJ, indicating in each case the intensity ratio of the band corresponding to each phosphorylated protein regarding to the corresponding actin bands.

### 2.9. Animal Experiment

(1)Ethics Statement for Use of Animals

Animal study and experiments were approved and performed under Jeonbuk National University (JBNU) Institutional Biosafety Committee (IBC, Protocol #:JBNU2021-08-001-001) and the Institutional Animal Care and Use Committee (IACUC, Protocol #:JBNU2022-05) in compliance with the Animal Welfare Act. Experiments were performed in BSL-3 Ag laboratory and facility in the Korea Zoonosis Research Institute, Jeonbuk National University in Iksan, Republic of Korea.

(2)Immunization and Challenge with ASFV Korean isolate

A total of 8 pigs (Sus scrofa) with healthy conditions and the same age (7 weeks old) were used for the experiment. The vaccinated group was composed of 4 pigs and the non-vaccinated group was composed by the other 4 pigs, which were added at the challenge. Additionally, another 6 five-week-old pigs were used for immunization without subsequent challenge to assess the attenuation. Experiments were performed in a BSL-3 Ag laboratory and facility. The pigs were acclimated for one week and managed with appropriate feeding and water supply system, cleaning and general veterinary care. Each experimental group was located in a separate pig isolator.

The vaccinated group was immunized with 10^2^ TCID50/pig vaccine virus via intramuscularly (IM). Whole blood and serum samples were collected on the days of vaccination. Four weeks after the immunization, pigs were challenged with 1 mL inoculum containing 10^2^ HAD_50_ of virulent ASFV, Korean isolate (ASF/Korea/Pig/Paju/2019, provided by APQA, Genebank Access No. MTT748042.1) IM.

(3)Samples collection and assessment of Clinical signs

Whole blood, nasal swab and fecal swab were collected at 0, 3, 5, 7, 10, 14, 21 and 28 days post vaccination (dpv) from vaccinated pigs and 0, 3, 5, 7, 10, 14, 21 days post challenge (dpc) from vaccinated group and control group pigs. Sera were collected at 0, 7, 10, 14, 21 and 28 dpv and 0, 5, 7, 14 and 21 dpc.

Virus detection for organ tissue samples was performed on pigs euthanized at 21 dpc or found dead after challenge. Tissues for virus detection were tonsil, lymph node (Mandibular, Superficial cervical, Gastrohepatic, Renal, Mesenteric), spleen, heart, lung, liver and kidney.

The clinical signs were evaluated and scored daily based on index previously established [51] (see Appendix A). Clinical signs included body temperature, inappetence, recumbency, skin hemorrhage, joint swelling, labored breathing, ocular discharge, diarrhea, urine and vomiting and was assigned a score with a maximum of 40. Euthanasia was performed if the accumulative clinical score was >20, i.e., when the pig had severe clinical signs. Gross lesion examination was performed on pigs euthanized or found dead after ASFV (Korean isolate) challenge. Gross lesions were scored following our previous study [52]. Briefly, gross lesion characterization and classification per organ was evaluated for body condition, integument, lung, liver, spleen, kidney, tonsil and lymph nodes.

### 2.10. ASFV Real-Time qPCR

Virus detection of whole blood swabs samples at different times post vaccination and post challenge and tissues from necropsy was determined by qPCR. ASFV DNA was isolated by DNeasy blood and tissue kit (Qiagen) according to the manufacturer’s procedure. Briefly, the sample volume for whole blood and swab samples was 100 µL and 200 µL, respectively. An amount of 100 µL of whole blood was mixed with 100 µL PBS, 20 µL of proteinase K, 4 µL RNase and 200 µL of Buffer AL. For swab samples, 200 µL swab samples were mixed with 20 µL of proteinase K, 4 µL RNase and 200 µL of Buffer AL. Vortex-mixed lysate was put into the column and centrifuge at 8000 rpm for 1 min and then wash with buffer AW1 and AW2 at 8000 rpm for 1 min. DNA was eluted using 200 µL of Buffer AE at 8000 rpm for 1 min after 3 min reaction at room temperature.

ASF specific real-time PCR was performed using a commercial kit (Vet maxTM African swine fever virus detection kit, ThermoFisher, Waltham, MA, USA) that is validated and certified by the OIE for ASFV detection. The reaction volume was 25 µL, 5 µL of sample DNA and 20 µL of reagents. The PCR condition was 50 °C for 2 min, 95 °C for 10 min and then 45 cycles of 95 °C, 15 s and 60 °C, 1 min. Results are expressed in Ct (cycle threshold), which is defined as the number of cycles required for the fluorescent signal to cross the threshold. Ct levels are inversely proportional to the amount of target nucleic acid in the sample.

### 2.11. ELISA and Neutralization Assay

To evaluate the humoral immune response, sera were collected according to the experiment design. ASF-specific antibodies were measured using commercial ELISA INgezim PPA COMPAC (Ingenasa, Madrid, Spain). The procedure was performed according to the manufacturer’s recommendation. Briefly, 50 µL of serum with equal volume (50 µL) of diluent was placed into a well and incubated for 60 min at 36 °C and then washed four times with washing solution. An amount of 100 µL of conjugate was added and incubated for 30 min at 36 °C and washed five times. Then, 100 µL of substrate was added and kept for 15 min at room temperature. After 100 µL of stop solution, the OD value at 450nm was measured. The positive blocking % was ≥50%. The neutralization assay was performed as explained in [53]. Briefly, sera from non-immunized (pre-immune) or immunized animals (from 28 dpv) were incubated at 56 °C for 30 min for inactivation, and then diluted 1/8 using DMEM supplemented with 20% of FBS and 0.05% Tween-80. Then, ASFV Arm-ΔCD2v-ΔA238L was incubated with either pre-immune or immune sera at 37 °C for 18 h, and then incubated virus was used to infect COS-1 cells (MOI = 0.1). At 72 hpi, supernatant containing the virus was titrated in COS-1 cells by using TCID-50 assay, as explained above.

## 3. Results

### 3.1. Generation of ASFV Recombinant Virus Arm-ΔCD2v-GFP-ΔA238L-mCherry by CRISPR-Cas9 Technology

A modified CRISPR-Cas9 technology in COS-1 cells was used in order to generate a new recombinant and attenuated virus with potential use as ASFV live attenuated vaccine (LAV). To achieve this, specific vectors detailed in Materials and Methods were transfected in COS-1 cells, which were then selected with puromycin and infected with wild-type virus virulent ASFV strain Arm/07/CBM/c2 (genotype II) in order to generate a single recombinant virus, Arm-ΔCD2v-GFP. Arm/07/CBM/c2 was isolated from Armenia07 strain stock, previously provided by the European Union reference laboratory for African swine fever (EURL-ASF) and INIA-CISA-CSIC (Madrid, Spain), by three plate purifications in COS-1 cells, as described in [23]. We previously verified that Arm/07/CBM/c2 was a highly virulent strain in animals’ experiment in which six pigs were inoculated with either 3 × 10^2^ HAD50 (Appendix A), or with 10^3^ HAD50 [54]. In both cases, all animals died between 7 and 10 dpi.

The EP402R gene, coding for CD2v protein, was replaced by the EGFP cassette, allowing the generation of the recombinant virus GFP (+) Arm-ΔCD2v-GFP. CD2v has been described as a gene with modulatory actions and a potential role in virulence in vivo [16,34,36,37,38,55]. Subsequently, COS-1-transfected cells were then transfected and infected with the single recombinant Arm-ΔCD2v-GFP, in order to generate the final double recombinant virus, Arm-ΔCD2v-GFP-ΔA238L-mCherry (Arm-ΔCD2v-ΔA238L), as detailed in the Material and Methods section. We targeted the A238L gene for deletion since we have largely described that A238L is involved in the control of several inflammatory precursors and immune response [25,28,31,32], thus presenting potential features to be deleted from a vaccine (Figure 1A).

### 3.2. Sequence Characterization of Recombinant Virus Arm-ΔCD2v-ΔA238L

Viral DNA from Arm-ΔCD2v-ΔA238L was extracted and sequenced by NGS, as explained in the Materials and Methods section. Illumina reads from Arm-ΔCD2v-ΔA238L were aligned against ASFV Arm/07/CBM/c2 reference sequence (accession no. LR812933). The estimated average coverage was calculated based on the percentage of mapped reads and genome size and resulted in 3698x. The high mean coverage obtained enables to unambiguously characterize the LAV prototype from a genetic point of view. In addition, as it can be observed in Figure 1B, mean coverage graph showed that all genome was cover up except for the deleted EP402R and A238L genes.

Genetic variability was analyzed by using sequencing data alignment as explained in the Materials and Methods. Numbers of single nucleotide polymorphisms (SNPs) and insertions/deletions (InDels) were determined and characterized by their location in coding and non-coding regions. Genetic variability analysis from ASFV Arm-ΔCD2v-ΔA238L showed a very low number of mutations (see Appendix A) when aligned against the ASFV Arm/07/CBM/c2 reference. We observed that most of the mutations (four indels) dropped into intergenic regions. One SNP and an indel provoked a frameshift in the MGF110-14L gene resulting in the addition of six amino acids at the C-terminal MGF110-14L. One SNP provoked a mutation in D205R gene, changing the Arg 205 to Gln and finally one indel provoked frameshift variant in the putative ORF ASFV G ACD 00190.

The de novo assembly of the genome of Arm-ΔCD2v-ΔA238L was performed, resulting in a 190626 bp length genome. We also confirmed the correct substitutions of the EP402R and A238L genes by the GFP and mCherry cassettes, respectively, by aligning the Arm-ΔCD2v-ΔA238L sequencing data against its theoretical mutant genome (data not shown), in silico generated for these purposes.

### 3.3. Growth of Arm-ΔCD2v-ΔA238L In Vitro Is Similar to the Growth of Wild-Type Arm/07/CBM/c2

The ability of Arm-ΔCD2v-ΔA238L to grow in porcine alveolar macrophages (PAMs) in vitro, compared to Arm/07/CBM/c2 wild-type virus was investigated. For that, PAMs were seeded at density of 1.6 × 10^6^ cells per well and infected with either Arm/07/CBM/c2 wild-type or Arm-ΔCD2v-ΔA238L at MOI = 0.5. As shown in Figure 2, in vitro growth of Arm-ΔCD2v-ΔA238L in PAM was similar to that of Arm/07/CBM/c2 wild-type, although slightly higher at all point post infection except at 72 hpi. However, these small increases could be due to differences also observed from 0 hpi. In any case, this result pointed us to hypothesize that the recombinant mutant would be able to efficiently replicate in PAMs during infection in pigs, to mount an effective immune response.

### 3.4. IFN Regulation In Vitro as a Signature of Attenuation of Arm-ΔCD2v-ΔA238L

Innate immune response is likely to be a key for host defense against viral pathogens. In order to evaluate the potential of Arm-ΔCD2v-ΔA238L as a suitable vaccine candidate, we have developed a new read-out assessing the virulence level through the IFN-I modulation in vitro. We have relied on our previous knowledge, as we have recently established that ASFV virulent strains efficiently block IFN-β production during PAM infection by preventing the activation of cGAS-STING pathway, whereas the natural attenuated strain NH/P68 does not [56]. Based on that, we tested the amount of IFN-β induced by our prototype in infected PAM comparing to the parental strain in vitro. PAMs were infected with either Arm/07/CBM/c2 or Arm-ΔCD2v-ΔA238L at MOI = 2 for 16 h, and IFN-β mRNA expression was evaluated by qPCR in each condition. As shown in Figure 3A, Arm-ΔCD2v-ΔA238L-infected PAMs expressed significantly higher amounts of IFN-β mRNA when compared to Arm/07/CBM/c2-infected PAMs. We also measured the mRNA quantity of the deleted genes CD2v (coded by EP402R) and A238L in both the recombinant and the wild-type virus, to verify the absence of expression of those genes in the recombinant virus. As controls of the infection, we quantify other viral proteins, such as p72 and p32, in order to verify that we are infecting with comparable amounts of the two viruses (Figure 3B).

Since we observed an increase in IFN-β mRNA levels in Arm-ΔCD2v-ΔA238L-infected PAMs compared to Arm/07/CBM/c2-infected PAMs, we further evaluated the activation of cGAS-STING pathway. First, we analyzed the activation of the pathway during the infection of PAMs with other attenuated or virulent strains, such as attenuated Uganda (Ug-AT) or E70, respectively. As can be seen in Figure 3C, at 16 hpi activation of two elements of this pathway, such as STING or TBK-1, occurs in PAMs infected with the attenuated strains (NH/P68 or Ug-AT), whereas it is completely inhibited in PAMs infected with the virulent strains (Arm/07/CBM/c2 or E70). Moreover, as shown in Figure 3D, several elements of cGAS-STING pathway were phosphorylated during the infection with Arm-ΔCD2v-ΔA238L. In particular, STING and TBK1 were phosphorylated at 4 hpi, and TBK1 also at 8 hpi, in Arm-ΔCD2v-ΔA238L-infected PAM unlike in Arm/07/CBM/c2-infected PAMs. Densitometry of the bands confirms activation in cells infected with Arm-ΔCD2v-ΔA238L compared to Arm/07/CBM/c2. We also observe a high increase in phosphorylation of IRF3 at 4, 8 and 16 hpi in Arm-ΔCD2v-ΔA238L-infected PAMs. The phosphorylation of these proteins implicates the activation of cGAS-STING pathways during Arm-ΔCD2v-ΔA238L-infected PAMs, according to what we observed with IFN-β production (Figure 3A), and in contrast to Arm/07/CBM/c2 wild-type. Finally, to further confirm pathway activation in Arm-ΔCD2v-ΔA238L-infected cells, we compared the phosphorylation of STING, TBK1 and IRF3 at 4 hpi in cells infected with Arm/07/CBM/c2, Arm-ΔCD2v-ΔA238L or the attenuated NH/P68 strain. As seen in Figure 3E, again activation is observed in both NH/P68-infected and Arm-ΔCD2v-ΔA238L-infected cells, whereas it is completely inhibited in Arm/07/CBM/c2-infected cells. Altogether, these results support that Arm-ΔCD2v-ΔA238L may have an attenuated phenotype in vivo and hence it could be a suitable candidate to be tested as a vaccine prototype.

### 3.5. Arm-ΔCD2v-ΔA238L Is Attenuated In Vivo

Based on the in vitro results pointing to the attenuated phenotype of Arm-ΔCD2v-ΔA238L, we evaluate whether the prototype could function as a new ASF vaccine, by performing a safety study in pigs.

A group of four seven-week-old pigs was inoculated IM with 10^2^ TCID50 per animal with Arm-ΔCD2v-ΔA238L. The animals were observed for 28 days for the development of ASF clinical signs. All pigs survived to vaccination and as it is shown in Figure 4, body temperature remained under standard values, physiologic normal range and no significant increase in clinical score index was seen throughout the observation period. In addition, we have performed another animal experiment in which we immunize a group of six five-week-old pigs IM with the same dose (10^2^ TCID50 per animal). Again, all animals survived after 21 days, showed no side effects and their temperature remained within physiological ranges (Appendix A). These results indicate that Arm-ΔCD2v-ΔA238L is attenuated in vivo and hence may be a suitable candidate for vaccine prototype in the tested conditions.

### 3.6. Arm-ΔCD2v-ΔA238L Protects against a Virulent Challenge of Genotype II ASFV Korean Paju Strain

After confirming the attenuated phenotype of the prototype, as described above, 28 days post vaccination (dpv) pigs were challenged IM with 10^2^ HAD_50_ ASFV Korean isolate Paju strain. After challenge, all the control pigs, non-vaccinated, died or were euthanized between 7- and 12-days post challenge (dpc) (35–38 dpv). In contrast, 100% of the vaccinated animals continued to be healthy and alive three weeks after the challenge (Figure 5A). Accordingly, body temperature in all vaccinated animals remained under normal values except for animal #57 and #58 that experimented punctual fever peaks at days two and eight, respectively, but then regained normal temperature. Moreover, clinical score based on symptoms explained above (see Appendix A), were always under normal values for the observation period of 21 days (Figure 5B,C). In addition, necropsy of 21 dpc revealed that the organs of vaccinated animals showed no lesions, with a very low gross lesion score (Appendix A). In contrast, the non-vaccinated control animals experienced high fever (>41 °C) and typical ASF clinical signs from 5 dpc until they were found dead or euthanized (Figure 5B,C). Accordingly, non-vaccinated animals were also found to have gross lesions compatible with ASF, with a high gross-lesion score (Appendix A). As an example, the image of gastro-hepatic lymph node of a non-vaccinated animal versus a vaccinated animal is shown (Appendix A). These results indicate that ArmΔCD2v-ΔA238L is a promising vaccine candidate against ASFV.

### 3.7. Viremia and Virus Shedding Assessment in the Vaccinated Animals

Negative or residual blood viremia values were found in vaccinated animals, both before and after the challenge, whereas, on the contrary, the viremia found in blood in non-vaccinated animals after the challenge was very high (Table 2).

Nevertheless, though the negative viremia values in blood were so low, and in order to analyze the vaccine shedding that could occur in vaccinated animals, the presence of the virus in oral and fecal exudates was analyzed. As shown in Table 3, and in parallel to the values obtained in blood, the presence of the virus was absent or very low in these biological samples from all vaccinated animals, in contrast to the non-vaccinated animals, which showed high levels of viremia.

In vaccinated animals, we did not detect the ASF virus in nasal or fecal swabs during vaccination period, except for very low levels in nose swabs at 14 dpv (animals #57 and #58) and 21 dpv (animal #56) (Table 3). In fecal swabs, low levels of virus were detected at 14 dpv (animal #58) and at 21 dpv (animals #55, #57, #58), and residual viremia at 7 dpv (animal #55), 14 dpv (animal #55) and 28 dpv (animal #58) (Table 3). After the challenge, only low levels of viremia were detected at 10 dpc (animals #55, #57 and #58) while the rest of the time points analyzed and animal #56 viremia levels were undetectable (Table 2); and only low levels of virus were found in fecal swabs in animals #58 (at 3, 5, 10 and 14 dpc) and #57 (0 and 14 dpc) (Table 3). As expected, a high-viremia level was detected in all non-vaccinated animals at both nasal and fecal swabs (Table 3).

Overall, these results indicate that the vaccine prevented viremia and virus shedding in most of the vaccinated animals and after the challenge in blood, nasal and fecal swabs, the levels of viremia detected were almost undetectable.

Finally, low levels of viruses were detected in the tissues analyzed in the vaccinated animals (at 21 dpc), compared to the non-vaccinated animals after challenge, where high levels of viraemia were found (see Table 4). Tonsils showed higher virus detection relatively over other tissues in all vaccinated pigs. The challenge control group also showed that tonsils have the highest level of virus detection.

### 3.8. Specific Antibodies against ASFV Are Detected in Vaccinated Animals before and after Challenge with ASFV Virulent Korean Paju Strain

In order to shed light on the protective mechanism induced by the LAV Arm-ΔCD2v-ΔA238L prototype, we measured by ELISA the amount of specific anti-ASFV antibody we detected in vaccinated and unvaccinated animals, before and after the challenge. As it is shown in Figure 6A,B, a high amount of specific anti-ASFV antibodies is detected in vaccinated animals from 14–21 dpv, reaching the highest level at 28 dpv, close to 100%, when challenge was inoculated. These high levels were maintained until the end of the experiment (49 dpv, 21 dpc). Positive rate was reached 100% at 21 dpi and kept until 21 dpc. However, less than 20% of anti-ASFV detected in non-vaccinated animals at 3 dpc (31 dpv) that were nearly not detected at 7 dpc (35 dpv). In addition, we used sera from animals immunized at 28dpv to perform a neutralization assay. As can be seen in Figure 6C, infection of COS-1 cells decreases when virus incubated with sera from immunized animals at 28 dpv was used, compared to infection with virus incubated with pre-immune sera. This result indicates that sera from animals at 28 dpv contain neutralizing antibodies against ASFV. These results may indicate a relationship between the presence of anti-ASFV antibodies and protection in this LAV model against ASFV.

## 4. Discussion

Since ASFV represents a serious threat to economies around the world, the development of safe and effective ASF vaccines is urgently required. Currently, the most realistic approach for the generation of effective ASFV vaccines is LAVs; however, so far, no LAV for ASFV has been licensed or shown to be compatible with industrial-scale production [10,12,14,16,18,20] except for the recently reported production of the ASFV-G-ΔI177L LAV [11], whose behavior in farms and impact on epidemic control in Vietnam is currently being analyzed. Therefore, there is a critical need for further safe and effective candidate ASFV vaccines together with cell lines suitable for the generation and production of new candidate ASFV vaccines that do not affect their genetic stability.

Briefly, there are two main types of LAV: those naturally attenuated or derived, and those derived from virulent strains. The former includes the genotype I strain NH/P68 [14,57], and more recently the genotype II strain Lv17/WB/Rie1 [55]. Despite offering protection against homologous [14,55] and heterologous [14] challenges, vaccination with these strains usually produces uncontrolled side effects. Attempts to reduce these undesirable effects by rational deletion of certain genes proved unsuccessful [14], making vaccines based on attenuated strains unusable for commercial vaccination at present.

On the other hand, there are many prototypes based on virulent strains of genotype I [16,19,20] or genotype II [11,12,13,15,17,18], this last being the one currently circulating. Among these, the vast majority have been developed by eliminating one or more genes by homologous recombination [11,12,14,15,16,17,18,19,20] or by CRISPR-Cas9 [10], as is the case in the present work. Beyond the technology used to obtain the recombinant virus, a fundamental aspect for production, is in which cells, primary cells or cell lines, the prototype has been developed and grown. In most cases, reported prototypes have been developed in primary cells, such as porcine macrophages or PBMCs [11,12,13,18,20]. Industrial scale-up in these cells poses a problem due to reproducibility, cost, possible pathogens from the animals used as a source of the cells and finally raises ethical issues regarding animal welfare. The cell lines used to generate ASFV LAVs include WSL [58], PIPEC [59] and COS-1 [14,16]. However, the use of cell lines carries the risk of adaptation of the virus, which can lose numerous genes, the paradigmatic case being the Ba71V strain, adapted to grow in Vero cells, which is no longer able to infect pigs [60,61]. Genetic alterations have been also recently described in PIPEC adaptation of a LAV [59]. In our lab, we have not observed deletions during generation and growth of Arm-ΔCD2v-ΔA238L in COS-1 cells, despite we having passed our vaccine more than 10 passages in these cells (data not shown). Finally, an important aspect for the description and for the study of the stability of the prototype is NGS, which involves analyzing the deletions generated, but also the absence of other deletions and not-expected mutations. Although in recent years the use of this technology for the description of LAVs has become widespread, certain prototypes are still not adequately analyzed from this perspective [12,19,20], or the mean coverage is too low. In our case, we have used extracellular viral particles for NGS sequencing [23], avoiding eukaryotic DNA contamination in the sample and thus obtain high mean coverage in Illumina sequencing. Our analysis revealed that no extra deletions were found and only three not-expected mutations were found: a SNP in D205R, an indel in ASFV G ACD 00190 (a barely studied ASFV gene that has been only reported to be expressed at low levels in PBMCs during infection [62]) and an indel in MGF110-14L. Regarding the latter gene, the fact that the mutation appears in a polyC site makes it difficult to determine at this point, either by Illumina or by subsequent Sanger sequencing, whether the insertion is of one or more cytosines or whether it is due to sequencing errors.

In regard to the genes we have deleted in our current prototype, they have been extensively studied by our lab from the last ten years [25,28,29,30,31,32], establishing their function in vitro, and the mechanisms by which they interacts with the infected cell, giving us a prediction of their behavior in vivo. This previous knowledge is guaranteed before proceeding with the deletion of the genes in terms of being able to predict their role in vivo. A238L is a protein we described to be involved in the control of inflammatory mediators and the immune system through the control of NFkB and NFAT [24,25,26,27]. Furthermore, we have implicated to A238L in the control of TNF, COX2 and iNOS, as well as the transcriptional co-activator p300 [28,29,30,31,32]. The deletion of A238L, however, did not affect the virulence of Malawi Lil-20/1 [63] and E70 [64] strains. A238L has also been deleted from the naturally attenuated strain NH/P68, with the intention of reducing the side effects that the natural strain produced in vaccinated animals. In this case, although this objective was achieved, the protective efficacy of the prototype was reduced [14]. Finally, A238L has recently been deleted from the Kenya-IX-1033 strain, which did result in a partial decrease in virulence in vivo [10].

CD2v is a protein involved in HAD [34,40] as well as in the interaction with host proteins [35,36] and has recently been described as having a role in IFN production [37]. CD2v, and primarily HAD, has been linked to virulence in vivo since many of the naturally attenuated isolates turn out to be non-HAD [55,57,65]. In fact, the deletion of CD2v from the genome of virulent strains, such as Ba71 [16] or Kenya-IX-1033 [38], totally or partially reduces their virulence, respectively. However, in other cases, the deletion of the gene coding for CD2v alone has not produced any reduction in virulence in vivo [40,41]. The underlying molecular reasons are unknown, although it is very likely that the genetic context involved in the different genotypes plays a decisive role. Nevertheless, and based in our studies ongoing, we considered that CD2v may play a role in virulence and therefore considered eliminating it for our vaccine prototype, in combination with A238L. Other vaccine prototypes have been generated by deleting CD2v in combination with other virulence-related genes. In some cases, CD2v is part of successful combinations in the generation of attenuated and/or protective prototypes [39,66], and even its deletion has resulted in improved safety or efficacy in the protection of the prototypes [12,67]. However, in other cases, its deletion has not contributed to an improvement in the safety or efficacy of the vaccine prototype [68,69]. Again, the reasons why the absence of CD2v may produce opposite effects in vivo depending on the viral strain and the vaccine prototype are currently unknown.

Nevertheless, as we have shown, the resulting prototype of the double deletion, Arm-ΔCD2v-ΔA238L, was subjected to in vitro tests to check its possible virulence, and its interaction with the innate immune system, supporting its suitability of the candidate prior to subject it to in vivo trials. The previous in vitro tests we have developed are appropriate due to the cost of in vivo experiments, the lack of BSL-3 animal facilities worldwide, as well as to limit the number of animals involved in the tests. Hence, in vitro tests should help limit trials to vaccine candidates that present an attenuated phenotype in vitro. With this goal in mind, and based on our previous experimental data [56], we used the production of IFN-β and the activation or not of the cGAS-STING pathway in PAMs infected by either Arm-ΔCD2v-ΔA238L or Arm/07/CBM/c2 wild-type as hallmark for virulence. The results indicated that Arm-ΔCD2v-ΔA238L behavior was closer to attenuated strains/prototypes. Subsequent in vivo data have proved this trial to be correct, as animals inoculated IM only with 10^2^ TCID50 survived and did not have any side-effects or fever during the entire vaccination period. This suggests that these in vitro assays, based on the cGAS-STING pathway modulation, could facilitate the prior selection of vaccine candidates to be tested in vivo. Though we have confirmed it by using other attenuated and virulent strains, however, more data would be required to finally validate this in vitro test. In any case, Arm-ΔCD2v-ΔA238L in parallel to the attenuated strains was not capable to control cGAS-STING activation and, therefore, the IFN-β production in vitro, suggesting that CD2v and/or A238L could be involved in some step of the control of this pathway.

The immunization with Arm-ΔCD2v-ΔA238L was found to be attenuated in five- and seven-week-old pigs at 10^2^ TCID50/animal. Additional safety tests are ongoing to further verify the attenuation of the vaccine candidate (such as pregnant sows and reversion-to-virulence studies). In addition, studies with higher doses are also ongoing to further verify the attenuation of the candidate, as other vaccine prototypes have been reported in which virulence increases at higher doses, whereas they kept attenuated at lower doses [17]. However, a compromise dose between safety and efficacy within the margins established by the regulator should be established. In addition to its safety profile, Arm-ΔCD2v-ΔA238L also conferred protection against the virulent challenge of the Korean Paju strain in 100% of cases (4/4). Consistent with the protection data, high levels of specific antibodies vs. ASFV that were partially neutralizing were observed after vaccination, prior to the challenge, which may have contributed to the observed protection against the virulent challenge in the vaccinated animals.

## 5. Conclusions

In conclusion, this new ASFV vaccine prototype has the ideal characteristics in terms of production, safety and efficacy to become an effective and safe vaccine against ASFV and contributes to the control of this devastating worldwide pandemic. Nevertheless, more safety and efficacy tests would be needed for future commercial vaccine production. In this regard, future experiments are already planned for industrial production and potential commercialization of this vaccine.

## 6. Patents

The results presented in this manuscript resulted in the patent PCT/ES2022/070527.

## Figures and Tables

**Figure 1 vaccines-10-01992-f001:**
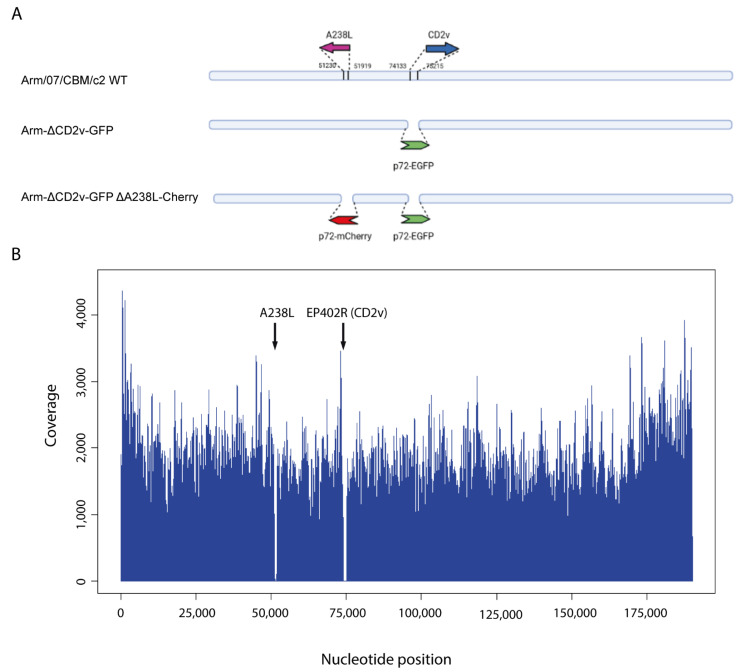
Generation of the recombinant Arm-ΔCD2v-ΔA238L. (**A**) Schematic for the generation of Arm-ΔCD2v-ΔA238L. Representation of the original Arm/07/CBM/c2 genome, the intermediate recombinant virus Arm-ΔCD2v-GFP generated by substitution of EP402R gene (coding for CD2v) with the EGFP gene under the control of the p72 promoter; and the final Arm-ΔCD2v-ΔA238L generated from the Arm-ΔCD2v-GFP recombinant virus by substitution of A238L gene with the mCherry gene under the control of the p72 promoter. (**B**) Whole genome coverage plot using Illumina reads of the recombinant Arm-ΔCD2v-ΔA238L virus mapped against ASFV Arm/07/CBM/c2 genome.

**Figure 2 vaccines-10-01992-f002:**
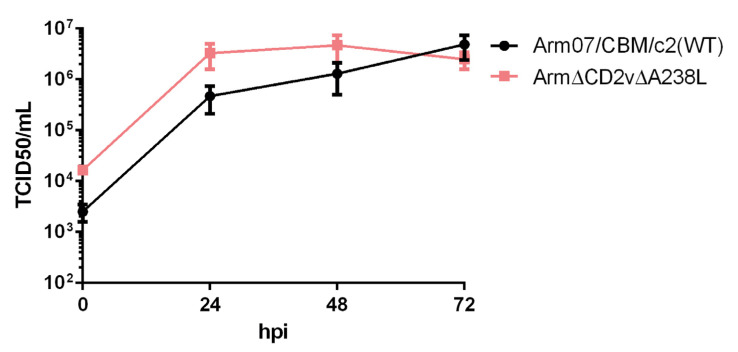
Arm/07/CBM/c2 and Arm-ΔCD2v-ΔA238L in vitro growth in PAM. PAM were infected with either Arm/07/CBM/c2 WT or Arm-ΔCD2v-ΔA238L at MOI = 0.5. At the indicated times post infection, total virus was recovered and titrated by TCID50 assay in COS-1 cells.

**Figure 3 vaccines-10-01992-f003:**
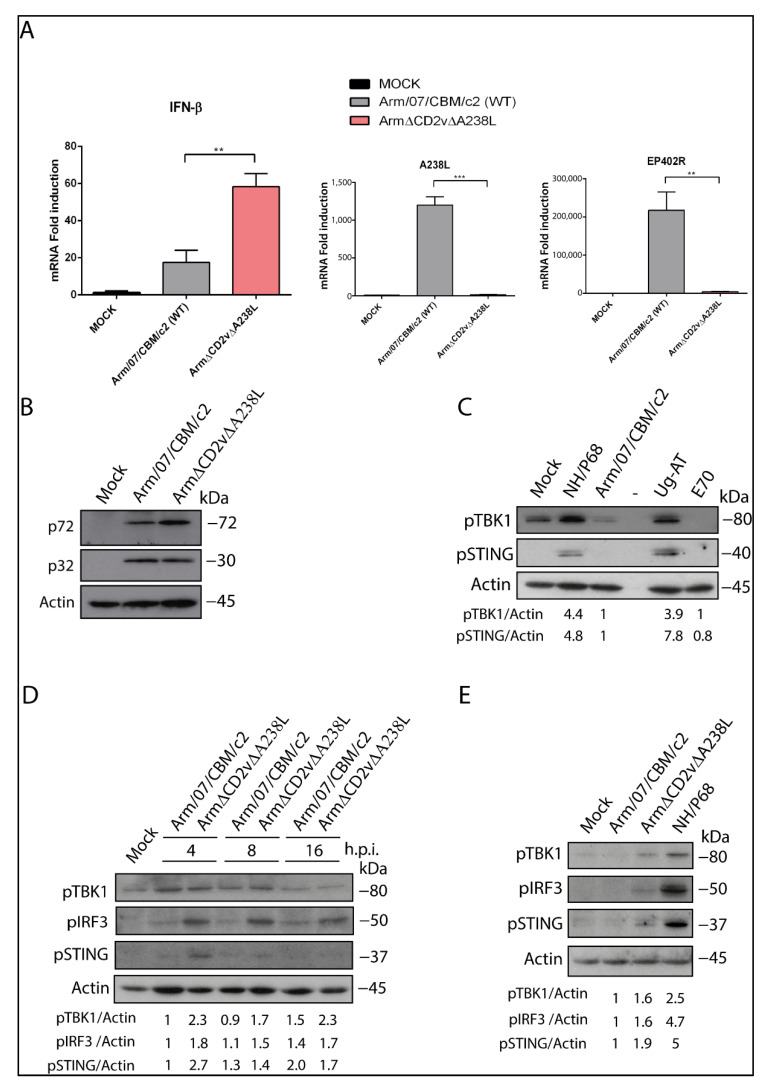
Arm-ΔCD2v-ΔA238L shows an attenuated phenotype in vitro. (**A**) IFN-β and viral genes EP402R and A238L mRNA was measured by qPCR in PAMs infected with either Arm-ΔCD2v-ΔA238L or Arm/07/CBM/c2 wild-type at 16 hpi, MOI = 3. (**B**) Viral protein expression was measured by Western blot. Cells were lysed in RIPA buffer at 16 hpi, MOI = 3, and lysates were separated by 7 to 20% SDS-PAGE, followed by immunoblotting with anti-ASFV p32, anti-ASFV p72 and anti-actin antibodies. (**C**–**E**) Activation of cGAS-STING pathway in PAMs infected with (**C**) attenuated (NH/P68 and Ug-AT) or virulent (Arm/07/CBM/c2 and E70) ASFV strains at 16 hpi; (**D**) Arm/07/CBM/c2 vs. Arm-ΔCD2v-ΔA238L at 4, 8 and 16 hpi; and (**E**) Arm/07/CBM/c2, Arm-ΔCD2v-ΔA238L and NH/P68 at 4 hpi. In all cases, the activation of cGAS-STING pathway was evaluated by phosphorylation status of cGAS, TBK1 and IRF3 by Western blot. Cells were lysed in RIPA buffer at 4, 8 and 16 hpi, MOI = 2, and lysates were separated by 7 to 20% SDS-PAGE, followed by immunoblotting with anti-pSTING, anti-pTBK1, anti-pIRF3 and anti-actin antibodies. **: *p*-value < 0.01; ***: *p*-value < 0.001. Two-way ANOVA-test performed with GraphPad.

**Figure 4 vaccines-10-01992-f004:**
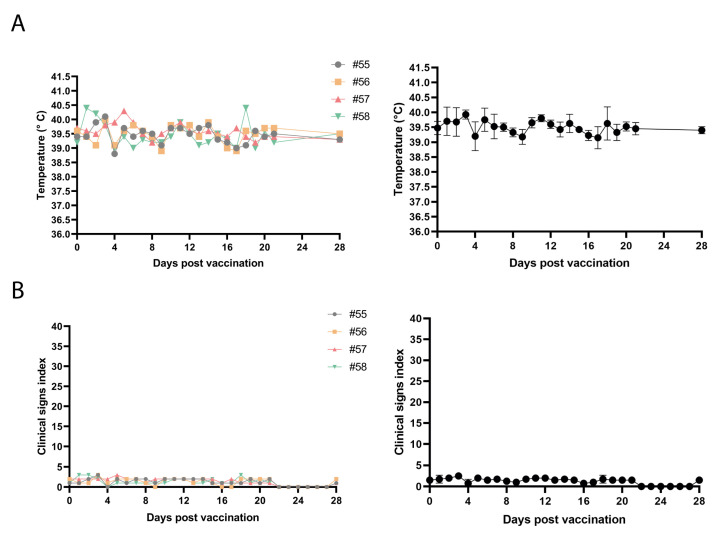
Clinical safety profile study of pigs inoculated with the vaccine prototype Arm-ΔCD2v-ΔA238L. (**A**) Individual (right) or average (left) daily temperatures of pigs receiving a 10^2^ TCID50 dose of the Arm-ΔCD2v-ΔA238L during 28 days after vaccination. (**B**) Clinical scores of individual pigs (**right**) or average (**left**) receiving a 10^2^ TCID50 dose of the Arm-ΔCD2v-ΔA238L during 28 days after vaccination. Scores were determined based on several parameters, including fever, inappetence, recumbency, skin hemorrhage, hemorrhagic areas on ears and body, joint swelling, labored breathing and/or coughing, ocular discharge, diarrhea, blood in urine and vomiting (see Appendix A).

**Figure 5 vaccines-10-01992-f005:**
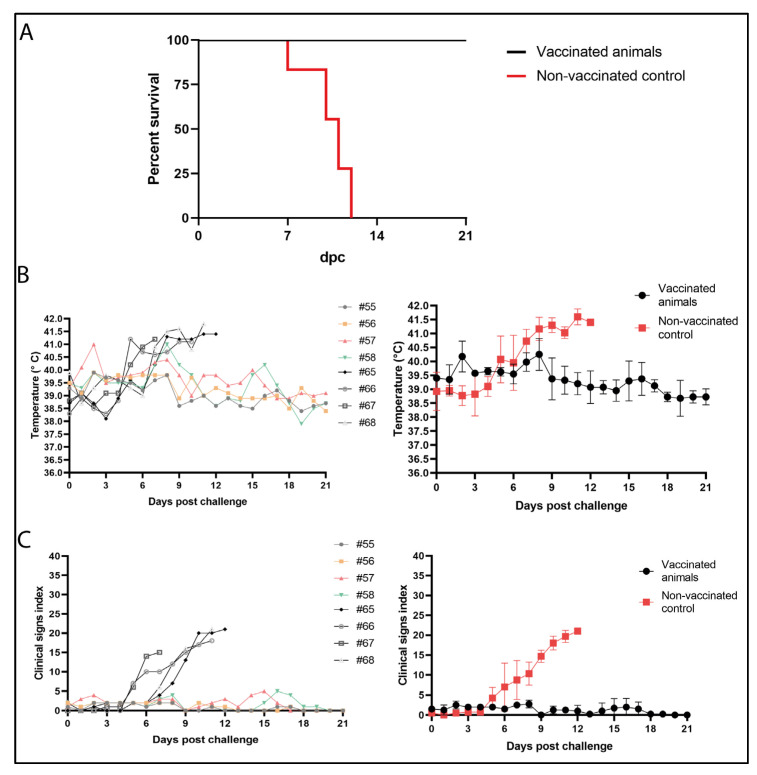
Clinical challenge of pigs inoculated with Arm-ΔCD2v-ΔA238L LAV candidate. (**A**) Survival analysis of vaccinated/challenged pigs (black) and non-vaccinated/challenged positive control pigs (red). (**B**) Individuals (**right**) or average (**left**) daily temperatures of pigs receiving a 10^2^ pfu dose of Arm-ΔCD2v-ΔA238L, followed after 28 days by challenge with 10^2^ HAD_50_ of virulent Korean Paju strain (animals #55–58; average in black); and non-vaccinated control pigs also challenged with virulent Korean Paju strain (animals #65–68, average in red). (**C**) Clinical scores of individual pigs (**right**) or average (**left**) during the challenge phase of vaccinated animals (animals #55–58; average in black) and non-vaccinated control pigs (animals #65–68, average in red). Scores were determined based on several parameters (see Appendix A).

**Figure 6 vaccines-10-01992-f006:**
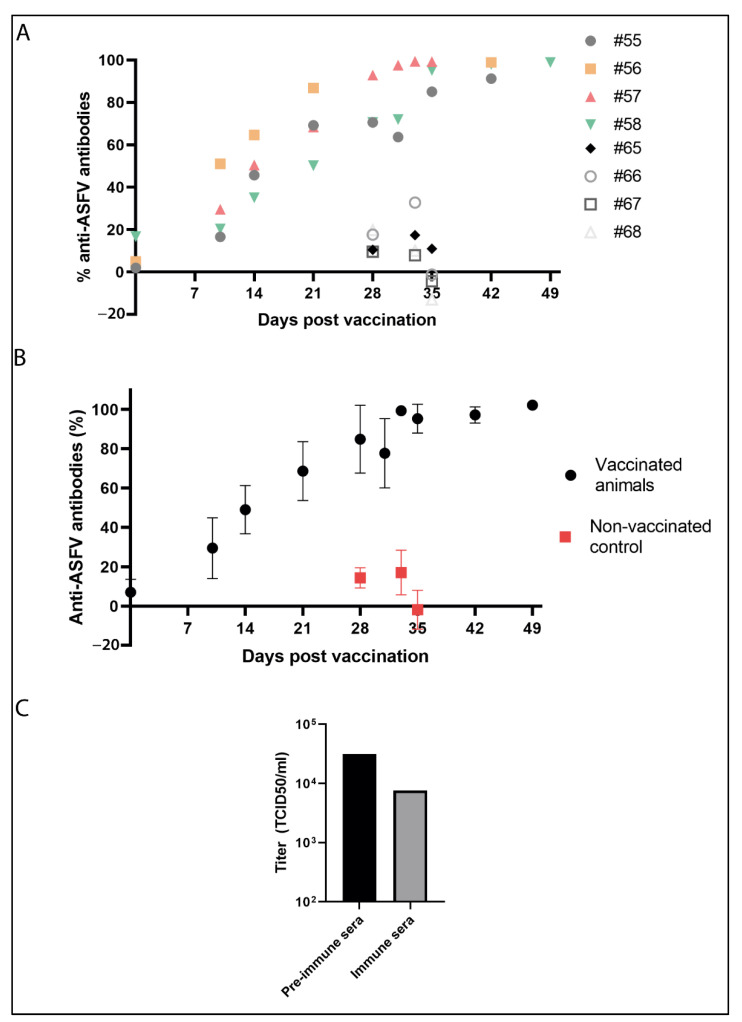
Specific anti-ASFV antibodies detected by ELISA can neutralize ASFV infection. ELISA (“ELISA INgezim PPA COMPAC” (Ingenasa), which uses p72 as viral protein) was used to quantify specific antibodies against ASFV in sera of individual (**A**) or mean (**B**) of vaccinated (#55–58) and non-vaccinated (#65–68) animals after vaccination. Challenge was inoculated at 28 dpv. More than 50% in considered positive for the presence of anti-ASFV antibodies. (**C**) Anti-ASFV antibodies partially neutralize ASFV infection. Arm-ΔCD2v-ΔA238L was preincubated with pre-immune or immune sera (28 dpv) and then used to infect COS-1 cells. 72 hpi supernatant containing virus was harvested and titrated in COS-1 cells.

**Table 1 vaccines-10-01992-t001:** Oligos used for the detection of recombinant and parental viruses by PCR.

Oligos to Detect:	
GFP (Recombinant virus)	5′-ACATGGTCCTGCTGGAGTTC-3′ 5′-GCCTGAAATACCAGAAAGAGAAGAC-3′
EP402R (Parental virus)	5′-CCATTAAGCATCATAATTGGGATAAC-3′ 5′-GCCTGAAATACCAGAAAGAGAAGAC-3′
mCherry (Recombinant virus)	5′-CATAACCAATTGCCCATCCTC-3′ 5′-AACTCCTTGATGATGGCCAT-3′
A238L (Parental virus)	5′-CATAACCAATTGCCCATCCTC-3′ 5′-GGAGTTAGTCAAATCTCTTAACCATG-3′

**Table 2 vaccines-10-01992-t002:** Viremia detected by qPCR in EDTA blood samples in vaccinated (#55–58) and non-vaccinated (#65–68) animals at the indicated days post vaccination (dpv) and days post challenge (dpc).

	Vaccinated Animals				Non-Vaccinated Animals				
dpv	#55	#56	#57	#58	#65	#66	#67	#68	
0	39.00	38.25	39.18	38.79	-	-	-	-	
3	ND	ND	ND	37.32	-	-	-	-	
5	ND	ND	ND	ND	-	-	-	-	
7	ND	39.15	ND	35.25	-	-	-	-	
10	37.08	39.81	36.69	41.10	-	-	-	-	
14	37.18	ND	38.05	ND	-	-	-	-	
21	ND	ND	ND	ND	-	-	-	-	
28	ND	ND	ND	ND	-	-	-	-	
dpc									Significance (vaccinated vs. non-vaccinated
0	ND	ND	ND	ND	ND	ND	ND	ND	-
3	ND	35.26	38.97	39.11	21.25	23.14	24.86	39.34	**
5	39.61	ND	37.17	37.30	20.41	21.86	22.42	36.56	***
7	ND	ND	37.83	40.18	19.91	21.32	22.28	22.98	****
10	36.28	ND	36.49	35.74			24.87	20.56	**
14	ND	ND	38.77	38.59					
21	ND	ND	35.60	38.83					

Challenge was administrated at 28 dpv. Values indicated Ct. ND = no detection of viral DNA. For significance calculation, ND values are considered Ct 40. **: *p*-value < 0.01; ***: *p*-value < 0.001; ****: *p*-value ≤ 0.0001. Two-way ANOVA-test performed with GraphPad.

**Table 3 vaccines-10-01992-t003:** Virus detected by qPCR in nasal and fecal swabs samples in vaccinated (#55–58) and non-vaccinated (#65–68) animals at the indicated days post vaccination (dpv) and days post challenge (dpc).

	Nasal Swabs			Fecal Swabs		
dpv	Vaccinated (Mean)	Non-Vaccinated (Mean)	Signifcance Vaccinate vs. Non-Vaccinated	Vaccinated (Media)	Non-Vaccinated (Media)	Signifcance
0	ND			ND		
3	ND			ND		
5	ND			ND		
7	ND			40 ± 0		
10	ND			ND		
14	38.2 ± 0.043			39.645 ± 0.5		
21	39.785 ± 0			39.08 ± 0.73		
28	ND			40 ± 0		
**dpc**						
0	ND	ND	-	ND	39.97 ± 0.065	-
3	ND	39.76 ± 0	-	38.22 ± 0	35.57 ± 2.28	-
5	ND	33.64 ± 2.44	*	38.22 ± 0	33.56 ± 5.58	-
7	ND	27.23 ± 4.03	****	ND	29.36 ± 0.8	**
10	39.73 ± 0.045	26.87 ± 4.29	****	39.15 ± 0	32.27 ± 2.87	*
14	ND	29.35 ± 2.96	****	38.71 ± 0.51	-	
21	ND	-		ND	-	

Challenge was administrated at 28 dpv. Values indicated Ct. ND = no detection of viral DNA. For significance calculation, ND values are considered Ct 40. *: *p*-value < 0.05; **: *p*-value < 0.01; ****: *p*-value ≤ 0.0001. Two-way ANOVA-test performed with GraphPad.

**Table 4 vaccines-10-01992-t004:** Viremia detected by qPCR in the indicated tissues in vaccinated (#55–58) and non-vaccinated (#65–68) animals at 21 dpc.

Samples	Vaccinated Animals				Non-Vaccinated Animals				Significance (Vaccinated vs. Non-Vaccinated)
	#55	#56	#57	#58	#65	#66	#67	#68	
Tonsil	33.99	36.03	24.03	33.22	16.75	18.04	18.96	18.48	**
Man * LN	36.59	ND	32.26	39.93	18.27	19.08	16.98	19.20	***
SC * LN	34.23	38.93	31.53	34.01	19.21	18.44	21.02	19.60	***
GH * LN	40.40	37.51	34.34	ND	18.28	21.37	19.33	22.15	***
Renal LN	38.11	ND	30.74	ND	18.98	21.45	18.46	20.74	**
Mes LN	30.99	ND	34.85	38.21	18.09	24.54	17.60	24.25	**
Spleen	ND	37.61	33.55	38.22	16.86	17.79	20.06	18.83	***
Lung	ND	ND	28.00	35.06	16.85	19.50	18.83	21.07	**
Heart	40.41	38.19	29.06	35.07	21.81	22.39	22..02	23.77	***
Liver	38.81	ND	33.08	38.07	16.44	18.92	18.69	18.07	***
Kidney	ND	40.13	34.56	35.55	21.06	22.16	22.09	23.61	***

Values indicated Ct. ND = no detection of viral DNA. * Man: Mandibular; SC: Superficial Cervical; GH: Gastrohepatic. **: *p*-value < 0.01; ***: *p*-value < 0.001.

## Data Availability

Not applicable.

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
