# Peer review of "Recombinant African Swine Fever Virus Arm/07/CBM/c2 Lacking CD2v and A238L Is Attenuated and Protects Pigs against Virulent Korean Paju Strain"

_vaccines, 2022, doi:10.3390/vaccines10121992_

Round 1

Reviewer 1 Report

The manuscript reported that the recombinant African swine fever virus lacking CD2v and A238L is attenuated and provided protection against the virulent challenge against the Korean Paju genotype II strain. It would provide an alternative candidate for the live attenuated vaccine development of ASFV. There are some concerns to be considered before accept for publication.

1.      Biosafety of the live attenuated vaccine (LAV) is the major concern for ASFV vaccine development. The current data from 4 pigs (7 weeks old) vaccinated with 102 TCID50/pig did not provide sufficient evidence for the safe consideration. It is suggested that pigs of different ages (such as piglet and sow) be infected with different dose of the recombinant virus and observed for a long duration. Furthermore, the possibility of mutation and recombination the Arm-ΔCD2v-ΔA238L virus in pigs should be analyzed and discussed.

2.      In the challenge experiments, the authors evaluated protective effect of Arm-ΔCD2v-ΔA23 against the virulent challenge by recording body temperature, mortality and testing viremia titers. However, the histopathology and/or immunohistochemistry data or pictures of the corresponding infected tissues were not found to provide sufficient evidence for the author’s conclusion.

3.      The data of qPCR for viremia detection in the Tables (Table 2-5) are suggested to recombine as one or two Tables to compare the virus titer in different samples after vaccination as well as challenge. In addition, The Ct values of qPCR from different animals of each group should be analyzed statistically. I do not understand the meaning of “39,00”, which should be explained in the Methods Section or the Table Note.

4.      In the Figure 5B and 5C, “Days post vaccination” in the left Pics should be “Days post challenge”?

5.      In the Figure 6, which viral protein of ASFV is used to testing for the specific anti-ASFV antibodies detected by ELISA?

Author Response

The manuscript reported that the recombinant African swine fever virus lacking CD2v and A238L is attenuated and provided protection against the virulent challenge against the Korean Paju genotype II strain. It would provide an alternative candidate for the live attenuated vaccine development of ASFV. There are some concerns to be considered before accept for publication.

We thank the reviewer for his/her comments.

  1. Biosafety of the live attenuated vaccine (LAV) is the major concern for ASFV vaccine development. The current data from 4 pigs (7 weeks old) vaccinated with 102 TCID50/pig did not provide sufficient evidence for the safe consideration. It is suggested that pigs of different ages (such as piglet and sow) be infected with different dose of the recombinant virus and observed for a long duration. Furthermore, the possibility of mutation and recombination the Arm-ΔCD2v-ΔA238L virus in pigs should be analyzed and discussed.

We agree with the reviewer that more experiments are needed in order to verify the complete attenuation of the vaccine prototype, although in the conditions tested the vaccine proved to be fully attenuated since no undesirable side effects were found. In order to clarify this point, we have included an explanation in Discussion (lines 774-782). In addition, we have added a new supplementary Figure (Figure S2) in which the safety of the prototype in 5-weeks old pigs is further shown (in lines 331-333; 535-539; 774-777). We are aware of the importance of the other experiments suggested by the Reviewer, such as safety studies with different doses, or the possibility of mutation and recombination in pigs (back-passage experiments). In fact, they are currently underway and we hope will be published in the near future. However, in the current work, we believe we are innovating beyond the state of the art in the development of vaccines against ASFV, since we have deleted a gene, A238L, which, as shown in successive papers from our group (Granja et al, see REFS), encodes a protein capable of regulating NFkB and NFAT, modulating the expression of several pro-inflammatory factors. We think that the double recombinant generated, and the fact of the resultant LAV is safe and protective against virulent challenge, would be interesting for the scientific community.

  1. In the challenge experiments, the authors evaluated protective effect of Arm-ΔCD2v-ΔA23 against the virulent challenge by recording body temperature, mortality and testing viremia titers. However, the histopathology and/or immunohistochemistry data or pictures of the corresponding infected tissues were not found to provide sufficient evidence for the author’s conclusion.

In order to improve the manuscript and following the suggestion of the reviewer, we have added now pictures showing the organs of the vaccinated animals vs. non-vaccinated after challenge, together with the gross lesion score, in the new Figure S3, including a description in lines 355-359 and 562-568.

  1. The data of qPCR for viremia detection in the Tables (Table 2-5) are suggested to recombine as one or two Tables to compare the virus titer in different samples after vaccination as well as challenge. In addition, The Ct values of qPCR from different animals of each group should be analyzed statistically. I do not understand the meaning of “39,00”, which should be explained in the Methods Section or the Table Note.

We have followed the suggestion of the reviewer and we have now fuse Tables 3-4 into new Table 3. Since the resulting table was too large, we have added only the mean and significance in the new Table 3, and we have added the previous Table 3 and Table 4 as supplementary tables Table S3 and Table S4. We have also included the statistically significance of the Ct values per group in Table 2, Table 5 and new Table 3. We also added a brief description of Ct meaning in Mat&Meth section (lines 376-379).

  1. In the Figure 5B and 5C, “Days post vaccination” in the left Pics should be “Days post challenge”?

Yes, the reviewer is right. We apologize for our mistake. We have now change it correctly to “Days post challenge”.

  1. In the Figure 6, which viral protein of ASFV is used to testing for the specific anti-ASFV antibodies detected by ELISA?

The ELISA was performed by the commercial kit “ELISA INgezim PPA COMPAC” (Ingenasa), which uses p72 as viral protein. We have added this information in the Figure legend for further clarification (lines 658-660).

Reviewer 2 Report

General comments

Núñez and colleagues used COS-1 cells and the CRISPR-Cas9 system to generate a double knockout ASFV strain (Arm-ΔCD2v-ΔA238L) from a virulent field isolate. In this study, Arm-ΔCD2v-ΔA238L showed no virulence in pigs, and pigs immunized with the strain were completely protected from challenge by the highly virulent strain.

Throughout, this paper has many inappropriate notations that should be corrected. For example, CO2 (CO2), in vitro (in vitro), in vivo (in vivo), 1’5·106 cells (1.5×106 cells), and so on. Please check and correct it properly.

Major comments

・This paper focuses on two things, “evaluation of Arm-ΔCD2v-ΔA238L for vaccine candidate” and “in vitro evaluation of pathogenicity of ASFV”. The former point is very interesting and contains a lot of data supporting the results. The latter point is also important for establishing better vaccine candidates as soon as possible, however, the data is insufficient. If the authors want to argue this point, they should also present data with several virulent and attenuated strains to demonstrate the usefulness of the in vitro test. Moreover, the authors note that phosphorylation of STING and TBK1 at 4 or 8 hpi is activated in Arm-ΔCD2v-ΔA238L infected PAMs compared to wild-type strains. However, some of the bands shown in Figure 3C (especially 8 hpi of STING and TBK1) are unclear about the difference from the wild type. If you want to mention this point, you should quantify the bands or use attenuated strain as a control.

・In the growth curve analysis in PAM, the authors note that Arm-ΔCD2v-ΔA238L had higher titers compared to the wild type at all time points except 72 hpi. At 0 hpi, however, there is already a difference in the amount of virus-infected. There are no differences in the growth potential between wild type and Arm-ΔCD2v-ΔA238L, although the mention above may be misleading in that the growth potential of Arm-ΔCD2v-ΔA238L is higher than wild type virus. I was amazed the Arm/07/CBM/c2 (WT) strain easily infects Cos-1. We have inoculated Cos-1 with the Armenia07 strain several times in our laboratory, but no observed virus growth. What is the history of the Arm/07/CBM/c2 strain? This strain may be a Cos-1-adapted strain. Therefore, the authors should mention these.

・The authors mention that Arm-ΔCD2v-ΔA238L was attenuated entirely, but the virulence of the parent strain (Arm/07/CBM/c2) is not well described. Is it highly virulent strain? Also, this study only examined virulence at a viral dose of 102 TCID50/head. It has been reported that some recombinant viruses remain pathogenic when the pigs are infected with high titers of the viruses (O’Donnell et al., J. Virol, 2015). Therefore, it should be mentioned that careful discussion is necessary to determine whether the virus is truly attenuated or not.

Minor comments

Lines 26, 588, 650: Delete space before the first characters.

Line 54: Describe what is the abbreviation of "LAV" here. Then, correct line 57. 

Line 84: I think it is "EP402R", not "EP420R".

Lines 120-121: The sentence, "For the design of ...", is inaccurate. For example, "We designed the gRNA based on the sequence of the ASFV strain Arm/07/CBM/c2 (LR812933.1)." 

Lines 130-132: The sentence, "For the generation of ...", contains typographical errors. For example, "[...] we first clone the EP402R gene and its flanking regions (500bp upstream and 500bp downstream) (73,362 - 75,716), [...]".

Line 156: "bp", not "pb".

Line 161: I think it is "pFL-A238L", not "pFL-EP402R".

Line 188: The abbreviation "dpi" is already described in section 2.2.

Line 217: Change "DNAsa I" to "DNAse I".

Line 223: Write "[...] for 3 minutes [...]".

Lines 245-248: Write "[...] the coding regions (SNPs and InDels), we amplified by [...]".

Line 274: Write "At 0, 24, 48 or 72 hours post-infection (hpi), the cells were [...]".

Lines 277-281: I think the authors measured viral titer by fluorescent protein expression for recombinant viruses without fixation. Also, for wild-type viruses, viral titer was measured by IFA with fixation and permeabilization. If yes, please describe it correctly.

Line 333: What is the meaning of the abbreviation "HAU"? Is it the same as "HAD50"?

Line 349 and 351: Change "RT-qPCR" to "qPCR". Because the author used a similar abbreviation in section 2.7. 

Line 415: Change "Table1" to "TableS1".

Figure 3: The author uses "Arm/07/CBM/c2" or "WT" in this figure. I recommend using either description.

Line 510: According to Figure 5, I think it is "12-days", not "10-days".

Line 513: According to Figure 5, I think it is "#57 and #58", not "#56 and #58".

Line 543: Delete "n" after "in".

Lines 548-550: The sentence, “[...], and residual virulence at 7dpv (animal #55), 14 dpv (animal #55) and 28 dpv (animal #58) (Table 3).” refer as residual virulence. However, I do not understand the virulence from this Table. 

Lines 595-599: These sentences repeat "1. Introduction"; authors should delete or rewrite the sentences.

Line 628: Check the sentence "In our hands" is correct.

Line 639: The authors mention that these mutations are derived from off-target, but what is the evidence for this, meaning that gRNA sequence homology, or something else, was observed?

Author Response

General comments

Núñez and colleagues used COS-1 cells and the CRISPR-Cas9 system to generate a double knockout ASFV strain (Arm-ΔCD2v-ΔA238L) from a virulent field isolate. In this study, Arm-ΔCD2v-ΔA238L showed no virulence in pigs, and pigs immunized with the strain were completely protected from challenge by the highly virulent strain.

 Throughout, this paper has many inappropriate notations that should be corrected. For example, CO2 (CO2), in vitro (in vitro), in vivo (in vivo), 1’5·106 cells (1.5×106 cells), and so on. Please check and correct it properly.

We apologize for these mistakes that are now being corrected in the present version.  

Major comments

・This paper focuses on two things, “evaluation of Arm-ΔCD2v-ΔA238L for vaccine candidate” and “in vitro evaluation of pathogenicity of ASFV”. The former point is very interesting and contains a lot of data supporting the results. The latter point is also important for establishing better vaccine candidates as soon as possible, however, the data is insufficient. If the authors want to argue this point, they should also present data with several virulent and attenuated strains to demonstrate the usefulness of the in vitro test. Moreover, the authors note that phosphorylation of STING and TBK1 at 4 or 8 hpi is activated in Arm-ΔCD2v-ΔA238L infected PAMs compared to wild-type strains. However, some of the bands shown in Figure 3C (especially 8 hpi of STING and TBK1) are unclear about the difference from the wild type. If you want to mention this point, you should quantify the bands or use attenuated strain as a control.

We really thank the reviewer for his/her comment and the interesting observations, which improve the quality of this manuscript.

Regarding the “second point” assessed in this study, “in vitro evaluation of pathogenicity of ASFV”, and although it was not our intention to extend the methodology to evaluate ASFV virulence in vitro, it is true that we did apply our knowledge about cGAS-STING pathway modulation (García-Belmonte, et al., 2019) to firstly evaluate our candidate in vitro.  

Nevertheless, in this new version we have added a panel to Figure 3 illustrating the control of the cGAS-STING pathway by two attenuated wild type strains (NH/P68 and attenuated Uganda) vs. two virulent strains (Arm/07/CBM/c2 and E70) (new Figure 3C). Again, the virulent strains control the activation of the cGAS/STING pathway while the attenuated ones do not, where increased phosphorylation of these elements is observed (new Fig. 3C). These results confirm our hypothesis of the validity of this in vitro test to help to predict virulence in vivo.

We have also quantified the bands shown in the previous Fig. 3C (now, Figure 3D) as suggested by the reviewer. Although we agree with the reviewer that the differences between the Arm-ΔCD2v-ΔA238L and wild-type at 8 hpi in terms of STING and TBK1 phosphorylation are very slight, they indeed exist, as indicated by quantification (see new Figure 3D). Moreover, following the reviewer’s suggestion, we have also included another new panel indicating the phosphorylation level of STING, TBK1 and IRF3 including the attenuated control (NH/P68-infected PAM) at 4hpi (since all these events are mostly early).  It can be observed that the phosphorylation level of these proteins in NH/P68-infected PAM, is found in Arm-ΔCD2v-ΔA238L-infected PAM, albeit with lower intensity of the showed bands (new Figure 3E). We have described the new information in lines 490-495, 498-500 and 504-509.

Altogether, it is likely that the in vitro screening could be a good indicator for ASFV virulence in vivo. However, more information regarding the behavior of naturally attenuated or virulent strains as well as vaccine prototypes is needed for the complete validation of this test (as we are now emphasized in lines 767-770), which will be hopefully published at a later date.

・In the growth curve analysis in PAM, the authors note that Arm-ΔCD2v-ΔA238L had higher titers compared to the wild type at all time points except 72 hpi. At 0 hpi, however, there is already a difference in the amount of virus-infected. There are no differences in the growth potential between wild type and Arm-ΔCD2v-ΔA238L, although the mention above may be misleading in that the growth potential of Arm-ΔCD2v-ΔA238L is higher than wild type virus. I was amazed the Arm/07/CBM/c2 (WT) strain easily infects Cos-1. We have inoculated Cos-1 with the Armenia07 strain several times in our laboratory, but no observed virus growth. What is the history of the Arm/07/CBM/c2 strain? This strain may be a Cos-1-adapted strain. Therefore, the authors should mention these.

We again thank the reviewer for the interesting discussion. Firstly, we successfully grow ASFV virulent isolates (E70, 26544/OG10, Arm/07/CBM/c2), in COS-1 cells. (It would be interesting to see the differences in the virus growth protocol, or the differences in COS-1 clone(s), which may possibly explain this divergence among laboratories). However, we would like to point out that the data presented here refer to growth in PAM, not in COS-1. Virus obtained in each indicated time post infection (in PAM) was then titrated in COS-1 cells by TCID50, as indicated in the Figure Legend 2.

Concerning Arm/07/CBM/c2, this clone was isolated from an Armenia07 stock originating from the European Union reference laboratory for African swine fever (EURL-ASF), INIA-CISA-CSIC (Madrid, Spain) that was amplified three times in PAM in our laboratory, before being analyzed by Illumina and discovering the presence of at least two viral populations in this stock. These populations were subsequently isolated in COS-1 cells by plate isolation and then subject again to NGS analysis. One of these viral populations was named as Arm/07/CBM/c2, which resulted in 99.997% identity with Georgia 2007/1 strain. This information was previously published in the referenced paper Pérez-Núñez, et al., 2020, and we have now added a slight description in lines 404-407.   

・The authors mention that Arm-ΔCD2v-ΔA238L was attenuated entirely, but the virulence of the parent strain (Arm/07/CBM/c2) is not well described. Is it highly virulent strain? Also, this study only examined virulence at a viral dose of 102 TCID50/head. It has been reported that some recombinant viruses remain pathogenic when the pigs are infected with high titers of the viruses (O’Donnell et al., J. Virol, 2015). Therefore, it should be mentioned that careful discussion is necessary to determine whether the virus is truly attenuated or not.

We agree with the reviewer that in this study we only examined virulence at 102 TCID50/head and that we should perform more safety studies in order to verify the completely attenuation of the prototype. We have now comment on that in the discussion (lines 775-780) and we have included the reference mentioned by the reviewer. In this regard, as we comment in this new version (lines 780-782), what is important is to establish a commitment between safety and efficacy, i.e. the described prototype fully protect at 102 dose. However, we have also included an extra study with six 5-weeks-old animals in which we also observed the attenuation of the prototype 21 days after immunization (Figure S2, lines 535-539).

By the other hand, we have recently published an in vivo experiment with Arm/07/CBM/c2 in which it is shown the virulent behavior of the strain (Walczak et al, 2022) which have been now referenced in the new version. In addition, we have added as Supplementary Figure S1 a previously in vivo experiment we performed in which we used Arm/07/CBM/c2, showing again the virulence of this strain. This information has been described in the new text in lines 407-410.

Minor comments

Lines 26, 588, 650: Delete space before the first characters. Done

Line 54: Describe what is the abbreviation of "LAV" here. Then, correct line 57. Done

Line 84: I think it is "EP402R", not "EP420R". Done

Lines 120-121: The sentence, "For the design of ...", is inaccurate. For example, "We designed the gRNA based on the sequence of the ASFV strain Arm/07/CBM/c2 (LR812933.1)." Done

Lines 130-132: The sentence, "For the generation of ...", contains typographical errors. For example, "[...] we first clone the EP402R gene and its flanking regions (500bp upstream and 500bp downstream) (73,362 - 75,716), [...]". Done

Line 156: "bp", not "pb". Done

Line 161: I think it is "pFL-A238L", not "pFL-EP402R". Done

Line 188: The abbreviation "dpi" is already described in section 2.2. Done

Line 217: Change "DNAsa I" to "DNAse I". Done

Line 223: Write "[...] for 3 minutes [...]". Done

Lines 245-248: Write "[...] the coding regions (SNPs and InDels), we amplified by [...]". Done

Line 274: Write "At 0, 24, 48 or 72 hours post-infection (hpi), the cells were [...]". Done

Lines 277-281: I think the authors measured viral titer by fluorescent protein expression for recombinant viruses without fixation. Also, for wild-type viruses, viral titer was measured by IFA with fixation and permeabilization. If yes, please describe it correctly. For titration we have fixed both recombinant and wild-type samples. However, we have only permeabilized the wild-type samples. We have now specified in lines 282,283, 286.  

Line 333: What is the meaning of the abbreviation "HAU"? Is it the same as "HAD50"? Yes, but for clarification we changed in the text HAU to HAD50.

Line 349 and 351: Change "RT-qPCR" to "qPCR". Because the author used a similar abbreviation in section 2.7. Done

Line 415: Change "Table1" to "TableS1". Done

Figure 3: The author uses "Arm/07/CBM/c2" or "WT" in this figure. I recommend using either description. We use now “Arm/07/CBM/c2” in all panels in the new Figure 3.

Line 510: According to Figure 5, I think it is "12-days", not "10-days". The reviewer is right, we apologize for the mistake. We have now corrected in the text.

Line 513: According to Figure 5, I think it is "#57 and #58", not "#56 and #58". Done

Line 543: Delete "n" after "in". Done

Lines 548-550: The sentence, “[...], and residual virulence at 7dpv (animal #55), 14 dpv (animal #55) and 28 dpv (animal #58) (Table 3).” refer as residual virulence. However, I do not understand the virulence from this Table. We have now changed “virulence” by “viremia”.

Lines 595-599: These sentences repeat "1. Introduction"; authors should delete or rewrite the sentences. We have now rewritten these first sentences of the Discussion to avoid repeating the introduction (lines 667-672).

Line 628: Check the sentence "In our hands" is correct. We have now changed "In our hands" to "In our lab".

Line 639: The authors mention that these mutations are derived from off-target, but what is the evidence for this, meaning that gRNA sequence homology, or something else, was observed? When we talked about “off-target mutations” we were referring to mutations that we did not expect to happen, outside the initial targets (A238L and EP402R). For clarification, we have now changed “off-target” to “not-expected” mutations.

Reviewer 3 Report

This manuscript described the development of vaccine against African swine fever (ASF). ASF is a serious contagious diseases in both wild and domestic pigs. Therefore, the disease has been received high concern in the pig raising countries. To control ASF, several strategies have been developed. One of those is the vaccine. However, there are several obstacles to be overcome in the development of vaccine because the ASF virus, the etiological agent of ASF, has very characteristic genetic features. To s olve the problem, authors tried to develop mutant of ASF virus and challenge the pig immunized with the mutant of ASF virus with a Korean ASF virus isolate, Paju strain. This kind of trial will be helpful to control the ASF. 

Generally, this study was well designed and carried out. But, the main weak point is the number of pigs used in this study. Authors immunized only four pigs with the experimental vaccine. 

Also, only four pigs were control. With the number of pigs, how can authors tell "fully protects p igs against virulent Korean Paju strain" as shown in the title of this manuscript? This study is the only preliminary. With the repeat of this kind of study, authors can tell “this recombinant ASF virus fully protects against virulent Korean Paju strain". Otherwise, authors can change the title of this manuscript up to development of mutant of ASF virus and preliminary animal test. Also, authors measured the ASF virus specific antibodies by ELISA. To protect the disease, neutralizing antibody is more import ant than antigen specific antibody. So, authors should demonstrate data on the ASF virus neutralizing antibody.

Author Response

This manuscript described the development of vaccine against African swine fever (ASF). ASF is a serious contagious diseases in both wild and domestic pigs. Therefore, the disease has been received high concern in the pig raising countries. To control ASF, several strategies have been developed. One of those is the vaccine. However, there are several obstacles to be overcome in the development of vaccine because the ASF virus, the etiological agent of ASF, has very characteristic genetic features. To s olve the problem, authors tried to develop mutant of ASF virus and challenge the pig immunized with the mutant of ASF virus with a Korean ASF virus isolate, Paju strain. This kind of trial will be helpful to control the ASF. 

 We thank the reviewer for his/her kind comment.

Generally, this study was well designed and carried out. But, the main weak point is the number of pigs used in this study. Authors immunized only four pigs with the experimental vaccine. Also, only four pigs were control.

We agree with the reviewer that only four animals were used in this study and that more studies are needed in order to bring this vaccine prototype to market. To amend this weakness, we have now included data for another animal study (Supplementary Figure S2) in which six animals of 5 weeks-old were vaccinated and did not present any side effect or adversary reactions (see lines 535-539), reinforcing the idea that the vaccine prototype is safe, even in younger animals. In the regard the number of controls used, we have added an extra-experiment in which six control pigs were inoculated with the ASFV virulent Arm/07/CBM/c2 strain (Supplementary Figure S1, lines 407-409). More safety and efficacy studies are now being carried out (lines 775-780) and will be presented at a later date.      

With the number of pigs, how can authors tell "fully protects p igs against virulent Korean Paju strain" as shown in the title of this manuscript? This study is the only preliminary. With the repeat of this kind of study, authors can tell “this recombinant ASF virus fully protects against virulent Korean Paju strain". Otherwise, authors can change the title of this manuscript up to development of mutant of ASF virus and preliminary animal test.

In agreement to the reviewer’s suggestion, we have now changed the title by removing the word "fully". We have also removed the word “fully” from “fully attenuated” in lines 526 and 539

Also, authors measured the ASF virus specific antibodies by ELISA. To protect the disease, neutralizing antibody is more import ant than antigen specific antibody. So, authors should demonstrate data on the ASF virus neutralizing antibody.

In order to improve the manuscript and following the reviewer’s advice, we have performed a neutralizing antibody detection experiment with the sera of vaccinated animals at 28 dpv. This experiment is now being added in New Figure 6 (Figure 6C) including the description in lines 380, 389-395, 650-654, 658-665 and 784.

Round 2

Reviewer 1 Report

No.

Author Response

We thank the reviewer for his/her positive evaluation.

Reviewer 2 Report

Thank you so much to the authors for including the information solicited. The author's corrections have greatly improved the manuscript. However, I could not find FigureS1, S2, and S3 in your revised manuscript. Moreover, throughout the revised manuscript, it is not clear which part the author corrected because the corrections did not correspond one-to-one with the reviewer's comments. Therefore, I cannot make an accurate judgment of your manuscript. I would like you to submit a revised manuscript that clarifies the corrections.

Major comments

1.The authors have not responded to the following comments. In the growth curve analysis in PAM, the authors note that Arm-ΔCD2v-ΔA238L had higher titers than the wild type at all time points except 72 hpi. At 0 hpi, however, there is already a difference in the amount of virus-infected. I thought it was likely that the difference in the initial infection dose contributed to this result. Therefore, I'm afraid I disagree with your interpretation of "the growth potential of Arm-ΔCD2v-ΔA238L is higher than wild-type virus". I would like some clarification or answer on this.

2.I didn't notice in the first revision, but the Materials and Methods did not include the method regarding the experiment on phosphorylation of TBK1. Please describe them.

Minor comments

・Line 95 and 693: You need to change "off-target" to "not-expected."

・Line 160: I think it is "A238L", not "A138L".

・Line 491-492: Phosphorylation levels of STING in 8 hpi seem to be almost equal in Arm-ΔCD2v-ΔA238L and Arm/07/CBM/c2. Thus, the sentence, "In particular, STING and TBK1..." needs to be corrected.

・Line 548, 570, 796, and 799: Change the abbreviation from "HAU" to "HAD50".

・Line 761: Remove improper spaces.

Author Response

Thank you so much to the authors for including the information solicited. The author's corrections have greatly improved the manuscript.

We thank the reviewer for his/her kind comment.

However, I could not find FigureS1, S2, and S3 in your revised manuscript.

We apologize for this mistake. We indeed upload all figures (including the supplementary Figures S1, S2 and S3) in the website for paper submission, but we did not include them in the word document for supplementary material. We have now included the supplementary Figures S1, S2 and S3 both in the website submission tool and in the word document for supplementary material.

Moreover, throughout the revised manuscript, it is not clear which part the author corrected because the corrections did not correspond one-to-one with the reviewer's comments. Therefore, I cannot make an accurate judgment of your manuscript. I would like you to submit a revised manuscript that clarifies the corrections.

We apologize again for that. In this new version we have highlighted in yellow the previous corrections indicated in the first evaluation by this reviewer, and we have highlighted in green the corrections indicated in the current evaluation for better understanding and clarity.  

 Major comments

1.The authors have not responded to the following comments. In the growth curve analysis in PAM, the authors note that Arm-ΔCD2v-ΔA238L had higher titers than the wild type at all time points except 72 hpi. At 0 hpi, however, there is already a difference in the amount of virus-infected. I thought it was likely that the difference in the initial infection dose contributed to this result. Therefore, I'm afraid I disagree with your interpretation of "the growth potential of Arm-ΔCD2v-ΔA238L is higher than wild-type virus". I would like some clarification or answer on this.

We agree with the reviewer. We have now added an explanation to this fact pointed out by the reviewer (“these small increases could be due to differences also observed from 0 hpi”), which we have highlighted in green for clarity, in lines 456-457.

I didn't notice in the first revision, but the Materials and Methods did not include the method regarding the experiment on phosphorylation of TBK1. Please describe them.

The reviewer is right. Now we have modified the title of WB Section, and we have added the experimental conditions and specific antibodies, corresponding to the panels in Figure 3 (Figure 3C, 3D and 3E), in addition to the methodology used for densitometry of bands. See lines 297, 299, 300, 307-308, 317-319.

 Minor comments

・Line 95 and 693: You need to change "off-target" to "not-expected." Done

・Line 160: I think it is "A238L", not "A138L". Done

・Line 491-492: Phosphorylation levels of STING in 8 hpi seem to be almost equal in Arm-ΔCD2v-ΔA238L and Arm/07/CBM/c2. Thus, the sentence, "In particular, STING and TBK1..." needs to be corrected. We have now corrected the sentence (line 497).

・Line 548, 570, 796, and 799: Change the abbreviation from "HAU" to "HAD50". Done

・Line 761: Remove improper spaces. Done